# On Margin-Based Cluster Recovery
# with Oracle Queries

**Marco Bressan**
Dept. of CS, Univ. of Milan, Italy
marco.bressan@unimi.it

**Nicolò Cesa-Bianchi**
DSRC & Dept. of CS, Univ. of Milan, Italy
nicolo.cesa-bianchi@unimi.it

**Silvio Lattanzi**
Google
silviol@google.com

**Andrea Paudice**
Dept. of CS, Univ. of Milan, Italy &
Istituto Italiano di Tecnologia, Italy
andrea.paudice@unimi.it

## Abstract

We study an active cluster recovery problem where, given a set of $n$ points and an oracle answering queries like "are these two points in the same cluster?", the task is to recover exactly all clusters using as few queries as possible. We begin by introducing a simple but general notion of margin between clusters that captures, as special cases, the margins used in previous works, the classic SVM margin, and standard notions of stability for center-based clusterings. Under our margin assumptions we design algorithms that, in a variety of settings, recover all clusters exactly using only $\mathcal{O}(\log n)$ queries. For $\mathbb{R}^m$, we give an algorithm that recovers *arbitrary* convex clusters, in polynomial time, and with a number of queries that is lower than the best existing algorithm by $\Theta(m^m)$ factors. For general pseudometric spaces, where clusters might not be convex or might not have any notion of shape, we give an algorithm that achieves the $\mathcal{O}(\log n)$ query bound, and is provably near-optimal as a function of the packing number of the space. Finally, for clusterings realized by binary concept classes, we give a combinatorial characterization of recoverability with $\mathcal{O}(\log n)$ queries, and we show that, for many concept classes in $\mathbb{R}^m$, this characterization is equivalent to our margin condition. Our results show a deep connection between cluster margins and active cluster recoverability.

## 1 Introduction

This work investigates the problem of exact cluster recovery using oracle queries, in the well-known framework introduced by Ashtiani et al. [2016]. We are given a set $X$ of $n$ points from some domain $\mathcal{X}$ (e.g., from the Euclidean $m$-dimensional space $\mathbb{R}^m$) and an oracle answering to same-cluster queries of the form "are these two points in the same cluster?" or label queries of the form "which cluster does this point belong to?". The oracle answers are consistent with some clustering $\mathcal{C} = (C_1, \ldots, C_k)$ of $X$ unknown to the algorithm, where $k$ is a fixed constant. The goal is to design an algorithm that recovers $\mathcal{C}$ by using as few queries as possible.

Clearly, if there are no restrictions on $\mathcal{C}$, then any algorithm needs $n$ queries in the worst case. Thus, our goal is to understand when $\mathcal{C}$ can be recovered efficiently; ideally, in polynomial time, and by making $\mathcal{O}(\log n)$ queries. A natural attempt is to consider clusterings with well-separated clusters, since this is what is often considered a "good clustering". The existing work goes precisely in this direction, starting with the following result of Ashtiani et al. [2016] for the case $X \subseteq \mathbb{R}^m$. If every cluster $C_i$ is separated from $X \setminus C_i$ by a ball centered in the center of mass of $C_i$, and that ball does not intersect other clusters even if expanded by a factor of $1 + \gamma$, then with high probability

$\text{poly}(k, m, 1/\gamma) \log n$ queries are sufficient to recover $\mathcal{C}$ in polynomial time. This is called "spherical margin" condition, and $\gamma$ is called the margin.

Unfortunately, the spherical margin condition is not very realistic, since it imposes a very precise shape upon the clusters. In a generalization attempt, Bressan et al. [2020] showed that one can actually recover clusters with ellipsoidal separators with arbitrary centers, by increasing the number of queries to $\text{poly}(k, m, 1/\gamma)(m/\gamma)^m \log n$. This result is achieved via boosted one-sided error learning, which works as follows. Suppose that, by making $f(k, m, \gamma)$ queries, we could identify correctly (with zero mistakes) a constant fraction of the points in some cluster $C_i$. Then, we could label those points as $i$, remove them from the dataset, and repeat. It is not hard to show that, after $\mathcal{O}(k \log n)$ rounds, we will have correctly labeled all the input points with high probability. The difficult task is, of course, to learn a constant fraction of some cluster $C_i$ with one-sided error (one-sided error means that any point of $X$ predicted to be in $C_i$ must be in $C_i$). The key insight in [Bressan et al., 2020] is that, if the clusters have margin $\gamma$ with respect to their ellipsoidal separators, then roughly $(m/\gamma)^m$ queries are sufficient. This leads to the following question: how much can this approach be extended?

In this work we provide several answers, revealing an interesting connection between margin-based cluster recovery and one-sided error learning. Our main contributions are as follows.

1. We introduce a new notion of margin in $\mathbb{R}^m$, that we call "convex hull margin" (Definition 2). This is a strict generalization of the margins of Bressan et al. [2020], Ashtiani et al. [2016] and of the usual SVM margin, and allows the clusters to have *any shape whatsoever* as long as they are convex. Under the convex hull margin, we develop a novel technique for learning with one-sided error that we call *convex hull expansion trick*. It essentially amounts to sampling many points from a single cluster and "inflate" their convex hull by a factor of $(1 + \gamma)$. This technique yields a polynomial-time exact cluster recovery algorithm that uses $\text{poly}(k, m, 1/\gamma)(1 + 1/\gamma)^m \log n$ queries (Theorem 1). The $(1 + 1/\gamma)^m$ dependence on $\gamma$ and $m$ is significantly better than that of Bressan et al. [2020], and closer to their lower bound of order $(1 + 1/\gamma)^{m/2}$.

2. We introduce a notion of cluster margin for general pseudometric spaces called *one-versus-all margin* (Definition 3). This notion of margin is strictly more general than convex hull margin, and captures, as special cases, standard notions of stability for clustering problems such as $k$-means or $k$-centers. We show that, if a clustering has one-versus-all margin, then it can be recovered with $M(\gamma) \text{poly}(k) \log n$ queries via a pure learning-theoretic approach (Theorem 3), where $M(\gamma)$ is a quantity related to the packing numbers of the pseudometric space. We show that the dependence on $M(\gamma)$ is essentially optimal, thus characterizing the recoverability of clusterings in this setting.

3. Finally, we show a connection between margin-based learning and exact active cluster recoverability, when clusters are realized by some concept class $\mathcal{H}$ (that is, when for each cluster $C_i$ there is a concept $h_i \in \mathcal{H}$ such that $X \cap h_i = C_i$). We show that if a certain combinatorial parameter, the *coslicing dimension* $\text{cosl}(\mathcal{H})$, is bounded, then one can learn clusterings with $\text{cosl}(\mathcal{H}) \text{poly}(k) \log n$ label queries; otherwise, $\Omega(n)$ queries are needed in the worst case (Theorem 4). Moreover we show that, for all concept classes in $\mathbb{R}^m$ that are closed under affine transformations and well-behaved in a natural sense, finite coslicing dimension and positive one-versus-all margin are equivalent (Theorem 5).

Note that actively learning a clustering is equivalent, up to a relabeling of the classes, to actively learning a multiclass classifier in the transductive realizable case. Hence our results apply to that case, too. In particular, our $\mathcal{O}(\log n)$ query bounds imply $\widetilde{\mathcal{O}}(\log 1/\varepsilon)$ query bounds for pool-based active learning [McCallum and Nigam, 1998] of multiclass classifiers, where $\varepsilon$ is the generalization error. To see this, draw a set $X$ of $\Theta(\varepsilon^{-1}(K \log 1/\varepsilon + \log 1/\delta))$ unlabeled samples from the underlying distribution, where $K$ is the relevant measure of capacity (e.g., the Natarajan dimension), run our algorithms over $X$, and compute a hypothesis consistent with the recovered labeling $\mathcal{C}$. These kinds of reductions are standard in active learning, see for instance [Kane et al., 2017].

On the other hand, our results *do not* apply to actively learning subsets $h \subseteq \mathcal{X}$ (that is, to active learning in the standard sense) if our margin conditions are only enjoyed by the set of positives $X_+ = X \cap h$. To see this, let $X \subset \mathbb{R}^2$ and suppose $X_+$ contains a single point. Then $X_+$ satisfies our conditions with unbounded margin w.r.t. the Euclidean distance, but any algorithm needs $\Omega(n)$ label queries to recover it. This does not happen with clustering because, in that case, every class enjoys the margin property, and in particular both the "positives" and the "negatives" for $k = 2$.

Table 1 compares the bounds resulting from the different notions of margins known.

| margin | query bound | reference |
|--------|-------------|-----------|
| spherical | $\mathcal{O}\left(k\log n + k^2 \frac{\log k + \log \frac{1}{\delta}}{\gamma^4}\right)$ | [Ashtiani et al., 2016] |
| ellipsoidal | $\mathcal{O}\left(k\log n \left(k^2 m^2 \log k + \max\left\{2^m, \mathcal{O}\left(\frac{m}{\gamma}\log\frac{m}{\gamma}\right)^m\right\}\right)\right)$ | [Bressan et al., 2020] |
| convex hull | $\mathcal{O}\left(k^3 m^5 \left(1 + \frac{1}{\gamma}\right)^m \log\left(1 + \frac{1}{\gamma}\right)\log n\right)$ | this work |
| one-versus-all | $\mathcal{O}(M^*(\gamma)\, k^2 \log k \log n)$ | this work |

Table 1: Summary of existing margin notions and corresponding known query bounds for same-cluster queries. The first three bounds assume $\mathcal{X} = \mathbb{R}^m$. The fourth bound is for general pseudometric spaces and $M^*(\gamma)$ is roughly a packing number—see Section 4. Note that $m$ can often be replaced by the maximum rank (i.e., the rank of the subspace spanned by the points) of any cluster, see [Bressan et al., 2020]. The spherical margin assumes, for every cluster $C_i$, that $\forall x \in C_i, y \in X \setminus C_i$, $d(y, \mu_i) > (1+\gamma)d(x, \mu_i)$, where $\mu_i = \frac{1}{|C_i|}\sum_{x \in C_i} x$. The ellipsoidal margin assumes that for some PSD matrix $W_i \in \mathbb{R}^{m \times m}$ and some $c_i \in R^m$, $\forall x \in C_i, y \in X \setminus C_i, d_W(y, c_i) > \sqrt{1+\gamma}d_W(x, c_i)$, where $d_W(a, b) = \sqrt{\langle a - b, W(a-b)\rangle}$; note that for $\gamma \ll 1$ this can be thought of as $d_W(y, c_i) > (1+\gamma)d_W(x, c_i)$.

**Related work.** Same-cluster queries were introduced formally in [Ashtiani et al., 2016] together with the active cluster recovery problem. Those queries are natural to implement in crowd-sourcing systems, and for this reason they have been extensively studied both in theory [Ailon et al., 2018a,b, Gamlath et al., 2018, Huleihel et al., 2019, Mazumdar and Pal, 2017, Mazumdar and Saha, 2017b,a, Saha and Subramanian, 2019, Vitale et al., 2019] and in practice [Firmani et al., 2018, Gruenheid et al., 2015, Verroios and Garcia-Molina, 2015, Verroios et al., 2017].

Various notions of margin are central in both active learning and cluster recovery [Xu et al., 2004, Balcan et al., 2007, Balcan and Long, 2013, Kane et al., 2017, Bressan et al., 2021]. Our coslicing dimension is similar to the slicing dimension of Kivinen [1995] and the star number of Hanneke and Yang [2015]. Our lower bounds, like many others, are inspired from a construction by Dasgupta [2004]. Our arguments based on packing numbers are similar to those based on the inference dimension of Kane et al. [2017] or the lossless sample compression of Hopkins et al. [2021], as we cannot infer the label of a point only when it is far from already-labeled points. A query bound similar to the one given by our convex hull expansion trick, but worse by a factor roughly $2^m$, can be inferred by adapting arguments of Hopkins et al. [2020b]. Combinatorial characterizations of multiclass learning have been proposed in the passive case by Ben-David et al. [1995], Rubinstein et al. [2009], Daniely and Shalev-Shwartz [2014]. Other learning settings related to one-sided and active learning are RPU learning [Rivest and Sloan, 1988] and perfect selective classification [El-Yaniv and Wiener, 2012] — see [Hopkins et al., 2020a] for a discussion.

## 2 Preliminaries and notation

All missing proofs can be found in the supplementary material. The input is a pair $(X, O)$, where $X$ is a set of $n$ points from some domain $\mathcal{X}$, and $O$ is a label oracle that, when queried on any $x \in X$, returns the cluster id $\mathcal{C}(x)$ of $x$. The oracle $O$ is consistent with a *latent clustering* $\mathcal{C} = (C_1, \ldots, C_k)$ of $X$, i.e., a $k$-tuple of pairwise disjoint sets whose union is $X$.[1] We allow clusters to be empty. Our goal is to recover $\mathcal{C}$ by making as few queries as possible to $O$. Queries can be made adaptively, that is, the $j$-th point to be queried can be chosen as a function of the answers to the first $j - 1$ queries. We express the number of queries as a function of $k$, $n$, and other parameters to be introduced later. This setting is essentially equivalent to the semi-supervised active clustering (SSAC) framework of Ashtiani et al. [2016], where the oracle answers same-cluster queries $\mathrm{SCQ}(x, y)$ that, for any two points $x, y \in X$, return TRUE iff $\mathcal{C}(x) = \mathcal{C}(y)$. We use label queries instead of SCQ queries for simplicity; any SCQ query can be emulated with two label queries. Conversely, the label of any point can be learned with $k$ SCQ queries, up to a relabeling of the clusters, so our bounds hold for an SCQ oracle as well if multiplied by $k$. For conciseness, we state query bounds in the form $f(k, m, n, \gamma)$,

---

[1]In line with previous works, we assume $k$ is fixed and known.

for instance $\mathcal{O}\big(k^2\,(1 + 1/\gamma)^m \log n\big)$. These bounds should be thought of as $\min\big\{n, f(k, m, n, \gamma)\big\}$, since obviously we never need to query the same point twice.

We often assume a metric or pseudometric $d$ over $\mathcal{X}$ (a pseudometric allows two distinct points to have distance 0). For $x \in \mathcal{X}$ and $r \geq 0$, the closed ball of radius $r$ centered at $x$ is $B(x, r) = \{y \in \mathcal{X} : d(x, y) \leq r\}$. For any $X \subset \mathcal{X}$, we denote by $\phi_d(X) = \sup_{x, x' \in X} d(x, x')$ the diameter of $X$ measured by $d$, and we define $\phi_d(\emptyset) = 0$. For any two sets $U, S \subset \mathcal{X}$, we denote by $d(U, S) = \inf_{x \in U, y \in S} d(U, S)$ their distance according to $d$, and we define $d(U, \emptyset) = \infty$. For any $X \subset \mathbb{R}^m$, we write $\mathrm{conv}(X)$ for the convex hull of $X$. The unit sphere in $\mathbb{R}^m$ is $S^{m-1} = \{x \in \mathbb{R}^m : \|x\|_2 = 1\}$. We recall some learning-theoretic facts. Let $\mathcal{H}$ be an arbitrary collection of subsets of $\mathcal{X}$ (i.e., a concept class). The intersection class of $\mathcal{H}$ is $I(\mathcal{H}) = \bigcup_{i \in \mathbb{N}}\{h_1 \cap \ldots \cap h_i : h_1, \ldots, h_i \in \mathcal{H}\}$. Given any $S \subset \mathcal{X}$ and any $S' \subseteq S$ realized by some $h^\star \in \mathcal{H}$, the smallest concept in $I(\mathcal{H})$ consistent with $S'$ is defined as $h^\circ = \bigcap\{h \in \mathcal{H} : h \cap S = S'\}$. Note that $h^\circ \subseteq h^\star$. Finally, we recall the definition of learning with one-sided error:

**Definition 1** (Kivinen [1995], Definition 4.4). *An algorithm $\mathcal{A}$ learns $\mathcal{H}$ with one-sided error $\varepsilon$ and confidence $\delta$ with $r$ examples if, for any target concept $h^\star \in \mathcal{H}$ and any probability measure $\mathcal{P}$ over $\mathcal{X}$, by drawing $r$ independent labeled examples from $\mathcal{P}$, the algorithm outputs a concept $h \subseteq h^\star$ such that $\mathcal{P}(h^\star \setminus h) \leq \varepsilon$ with probability at least $1 - \delta$.*

## 3 Margin-based exact recovery of clusters in $\mathbb{R}^m$

In this section we consider the case $\mathcal{X} = \mathbb{R}^m$. We show that the ellipsoidal margin of Bressan et al. [2020] can be significantly generalized, while retaining the $\mathcal{O}(\log n)$ query complexity, by introducing what we call the *convex hull margin*. In a nutshell the convex hull margin says that, given any cluster $C$, any point not in $C$ is separated by the convex hull of $C$ by a distance at least $\gamma$ times the diameter of $C$. Instead of using the Euclidean metric, however, we allow distances to be measured by *any* pseudometric over $\mathbb{R}^m$, which we do not need to know, and which may even differ from cluster to cluster. The only requirement is that the pseudometric be homogeneous and invariant under translation (i.e., induced by a seminorm).

**Definition 2** (Convex hull margin). *Let $D$ be the family of all pseudometrics induced by the seminorms over $\mathbb{R}^m$, and let $X \subset \mathbb{R}^m$ be a finite set. A clustering $\mathcal{C} = (C_1, \ldots, C_k)$ of $X$ has convex hull margin $\gamma$ if for every $i \in [k]$ there exists $d_i \in D$ such that:*

$$d_i\big(X \setminus C_i, \mathrm{conv}(C_i)\big) > \gamma\,\phi_{d_i}(C_i) \tag{1}$$

This definition has a few interesting properties. First, it strictly generalizes the ellipsoidal margin of Bressan et al. [2020] and the spherical margin of Ashtiani et al. [2016]. To see this, let $D$ be the class of all pseudometrics over $\mathbb{R}^m$ that can be written as $d_W(x, y) = \langle x - y, W(x - y) \rangle$ for some positive semidefinite matrix $W \in \mathbb{R}^{m \times m}$ (for the spherical margin, take $W = rI$ where $I$ is the identity matrix). Second, it strictly generalizes (the multiplicative version of) the classic SVM margin. Indeed, under the Euclidean metric, if $X$ has diameter $R = \phi(X)$ and every cluster $C$ can be separated from $X \setminus C$ by a linear separator whose boundary has distance $\rho$ from $X$, then $C$ has convex hull margin at least $\frac{\rho}{R}$; and there are cases with arbitrarily small (multiplicative) SVM margin but arbitrarily large convex hull margin, see Section 4. Finally, pseudometrics are versatile and can express, for instance, the Euclidean distance between points after a projection on a subspace. This models scenarios where each cluster only "cares" about a certain subset of the features.

Under the convex hull margin, we give a polynomial-time algorithm, named CHEATREC (for Convex Hull ExpAnsion Trick Recovery) that recovers $\mathcal{C}$ using $\mathcal{O}(\log n)$ queries.

**Theorem 1.** *Let $(X, O)$ be an instance whose latent clustering $\mathcal{C}$ has convex hull margin $\gamma > 0$. Then CHEATREC$(X, O, \gamma)$ outputs $\mathcal{C}$, runs in time $\mathrm{poly}(k, n, m)$, and with high probability makes a number of label queries to $O$ bounded by $\mathcal{O}\big(k^2 m^5 \,(1 + 1/\gamma)^m \log(1 + 1/\gamma) \log n\big)$.*

To put this result in perspective, consider the algorithm of Bressan et al. [2020]. Under an ellipsoidal margin of $\gamma_{EL}$, that algorithm achieves a query bound of roughly $\big(\frac{m}{\gamma_{EL}}\big)^m \log n$. One can check that an ellipsoidal margin of $\gamma_{EL}$ implies $\gamma \geq \frac{\gamma_{EL}}{3}$ for all $\gamma_{EL} \leq 1$[2]. Hence, in this range, our dependence on $\gamma$ is better by $\Theta(m)^m$ factors.

---

[2]Their definition uses squared distances, so the relationship with our margin is $1 + \gamma \geq \sqrt{1 + \gamma_{EL}}$.

Both CHEATREC and the algorithm of [Bressan et al., 2020] are based on boosting learners with one-sided error. What makes CHEATREC different is, thus, how it learns with one-sided error. To explain this, let us recall how algorithm of Bressan et al. [2020] works. The algorithm starts by drawing a uniform random sample of $\Theta(m^2)$ points $S_C$ from some cluster $C$. (As there are at most $k$ clusters, to draw $s$ points u.a.r. from some cluster we can just draw $ks$ points u.a.r. from $X$, query their labels, and take the subset of the majority label). Then, it computes an outer approximation of $\mathrm{conv}(S_C)$ by fitting a minimum-volume ellipsoid $E(S_C)$ to $S_C$. Since the VC dimension of $m$-dimensional ellipsoids is $O(m^2)$, by standard PAC bounds $E(S_C)$ contains half of $C$ with good probability. The problem is that $E(S_C)$ approximates $\mathrm{conv}(S_C)$ very roughly, and thus it may intersect a large portion of $X \setminus C$. Indeed, the algorithm makes roughly $(m/\gamma_{EL})^m$ additional queries to filter out all points in $E(S_C) \cap (X \setminus C)$.

This gives us the following idea: instead of spending that many queries on "cleaning" $E(S_C)$, we spend them to build a much larger sample $S_C$, so that we can approximate $\mathrm{conv}(S_C)$ using a class of convex bodies of VC dimension much higher than $\Theta(m^2)$ — for instance, a polytope on $(1 + 1/\gamma)^m$ vertices. That body should enclose $\mathrm{conv}(S_C)$ quite tightly; perhaps so tightly to contain no point outside $C$. This intuition leads to our technique, called *convex hull expansion trick*, which essentially amounts to drawing a large sample from some cluster $C$ and inflate its convex hull by a factor $1 + \gamma$. This technique yields bounds that, compared to [Bressan et al., 2020], are strictly better and hold under more general assumptions. In the rest of the section, we describe this trick and sketch the proof of Theorem 1. For the complete proof, see the supplementary material.

## 3.1 CHEATREC and the convex hull expansion trick

The starting point of CHEATREC is the following idea. Let $C$ be any cluster of $\mathcal{C}$, and let $S_C$ be any subset of $C$. Let $K = \mathrm{conv}(S_C)$, choose any $z \in K$, and let $Q = (1+\gamma)K$ be the scaling of $K$ with respect to $z$. Finally, let $d \in D$ be the pseudometric under which $C$ has convex hull margin $\gamma$. Since $d$ is homogeneous and invariant under translation (recall that every $d \in D$ is induced by a seminorm), any $y \in X$ such that $d(y, K) \leq \gamma\phi_d(K)$ must belong to $C$ as well. Therefore, $X \cap Q \subseteq C$. That is, $Q$ contains only points of $C$.

Clearly, knowing that $X \cap Q \subseteq C$ is not enough; in order to make progress, we must guarantee that $X \cap Q$ forms a good fraction of $C$ (in principle $X \cap Q$ could just coincide with $S_C$, in which case we learn nothing). To this end, we draw a uniform random sample $S$ from $X$ of size $|S| = ks$, where $s$ will be set later, and query all labels of $S$. For every cluster $C$ we let $S_C = S \cap C$; as there are at most $k$ clusters, at least one of them satisfies $|S_C| \geq s$. The convex hull expansion trick says that, if (1) $s$ is roughly $(1 + 1/\gamma)^m$, and (2) $z = \mu_K$, where $\mu_K$ is the center of mass of $K$, then $Q$ contains a good fraction of $C$ with good probability. This gives the desired query bounds for learning $C$ with one-sided error; to implement the trick in polynomial time, however, we have to replace $\mu_K$ with an approximation, since computing it exactly is hard [Rademacher, 2007]. The convex hull expansion trick in fact states that the claim about $Q$ holds even with such an approximation.

We give below a formal statement of the trick, and a sketch of its proof. The uniform probability measure $\mathcal{U}$ over $K$ is defined by $\mathcal{U}(K') = \frac{\mathrm{vol}(K')}{\mathrm{vol}(K)}$ for all measurable $K' \subseteq K$. A probability measure $\mathcal{P}$ is $\varepsilon$-uniform if $|\mathcal{P}(K') - \mathcal{U}(K')| \leq \varepsilon$ for all measurable $K' \subseteq K$.

**Lemma 1** (Convex hull expansion trick). *Fix $\gamma > 0$, and let $s = \Theta\big(m^5 \left(1 + 1/\gamma\right)^m \log \left(1 + 1/\gamma\right)\big)$ large enough. Let $S_C$ be a sample of $s$ independent uniform random points from some cluster $C$, and let $K = \mathrm{conv}(S_C)$. Let $X_1, \ldots, X_N$ be independent random points sampled $\varepsilon$-uniformly from $K$, with $\varepsilon \in \Theta(m^{-1})$ small enough and $N \in \Theta(m^2)$ large enough, and let $z = \frac{1}{N}\sum_{i=1}^N X_i$. Finally, let $Q = (1+\gamma)K$ where the center of the scaling is $z$. Then $\mathbb{P}\big(|Q \cap C| \geq |C|/2\big) \geq 1/2$.*

*Sketch of the proof.* For simplicity we ignore factors $\mathrm{poly}(m)$ and $\log(1/\gamma)$. To begin, suppose $Q$ was obtained by scaling $K$ about its own center of mass, $\mu_K$. By a probabilistic argument adapted from [Naszódi, 2018], there exists a polytope $P$ on roughly $(1 + 1/\gamma)^m$ vertices such that $K \subseteq P \subseteq Q$. Now, the class $\mathcal{P}_{t,m}$ of all polytopes on at most $t$ vertices in $\mathbb{R}^m$ has VC-dimension roughly $t$, see [Kupavskii, 2020]. Let then $t = (1 + 1/\gamma)^m$. Since $P \in \mathcal{P}_{t,m}$, and since $P$ is consistent with $S_C$ (that is, $S_C \subseteq P$), by standard PAC bounds $\mathbb{P}(|P \cap X| \geq |C|/2) \geq 1/2$ as long as $|S_C| = \Omega(t) = \Omega\big((1 + 1/\gamma)^m\big)$. But $P \subseteq Q$, so $\mathbb{P}(|Q \cap X| \geq |C|/2) \geq 1/2$ as well.

Now suppose that, in place of $\mu_K$, we use $z = \frac{1}{N}\sum_{i=1}^N X_i$ where each $X_i$ is an independent random points sampled $\varepsilon$-uniformly from $K$. We consider $K$ is in isotropic position. The radius of $K$ is then at most $m$, so $\|X_i\|_2 \le m$. By standard calculations this implies that, if $N = \Theta(m^2)$ and each $X_i$ comes from a distribution that is $\Theta(1/m)$-uniform over $K$, with good probability $z$ is at distance $\eta = O(1)$ from $\mu_K$. In particular, by increasing $N$ by constant factors, we can make arbitrarily small the probability that $\eta \le 1/3$. This is sufficient to adapt a result of Naszódi [2018], through an extension of Grunbaum's inequality for convex bodies due to Bertsimas and Vempala [2004], and show that a polytope $P$ such as the one described above still exists, even if we expand $K$ about $z$. $\quad\square$

We conclude with a note on implementing CHEATREC in polynomial time. First, we should sample the points $X_i$ efficiently. To this end, we transform $S_C$ so that $\mathrm{conv}(S_C)$ is in near-isotropic position. This amounts to computing John's ellipsoid for $S_C$, and applying the map that turns that ellipsoid into a ball, which takes polynomial time. Afterwards, we can draw every $X_i$ via the "hit-and-run from a corner" algorithm of Lovász and Vempala [2006], in time $\mathrm{poly}(m, |S_C|)\log\frac{m}{\varepsilon}$ — this includes the time to solve a linear program to determine when the random walk of [Lovász and Vempala, 2006] hits the boundary of $K$. Once we have $z = \frac{1}{N}\sum_{i=1}^N X_i$, the only remaining issue is to avoid computing $K = \mathrm{conv}(S_C)$ and $Q = (1 + \gamma)K$ explicitly as intersections of halfspaces, as this could take time $|S_C|^{\Theta(m)}$. Instead, we just rescale the sample $S_C$ about $z$, obtaining a representation of $Q$ as a set of vertices. Then, computing $X \cap Q$ amounts to solving for every $x \in X$ a feasibility problem, which takes polynomial time.

## 4 The one-versus-all margin

In this section, we let $\mathcal{X}$ be a generic space equipped with a set of pseudometrics. Like in Section 3, we want to formulate a notion of margin between clusters. However, we cannot express the margin in terms of diameter of convex hulls (since $\mathcal{X}$ need not be a vector space and may have no notion of convexity at all). Thus, we introduce a notion called *one-versus-all margin*. We prove that clusterings with positive one-versus-all margin can be recovered with $\mathcal{O}(\log n)$ oracle queries. However, we do not provide a running time analysis; the implementation of our algorithm depends on $\mathcal{X}$ and on the specific class of clusterings, and in general may take time superpolynomial in the size of the input. Let us introduce the one-versus-all margin.

**Definition 3** (One-versus-all margin). *Fix $k$ pseudometrics $d_1, \ldots, d_k$ over $\mathcal{X}$. A clustering $\mathcal{C} = (C_1, \ldots, C_k)$ of a finite set $X \subset \mathcal{X}$ has one-versus-all margin $\gamma$ with respect to $d_1, \ldots, d_k$ if for all $i \in [k]$ we have $d_i(X \setminus C_i, C_i) > \gamma\,\phi_{d_i}(C_i)$.*

**Remark.** Unlike the convex hull margin (Definition 2), here $d_1, \ldots, d_k$ are fixed in advance. The reason is that here $\mathcal{X}$ may not be a linear space, $C_1, \ldots, C_k$ may not be convex, or $d_1, \ldots, d_k$ may not be induced by seminorms. Hence, we cannot apply techniques like the convex hull expansion trick, which work *for every pseudometric* including $d_1, \ldots, d_k$. As a consequence, it is unclear how one can recover $\mathcal{C}$ efficiently without knowing $d_1, \ldots, d_k$ in advance.

Before continuing, we shall prove that the one-versus-all margin can be arbitrarily large while the multiplicative SVM margin is arbitrarily small. This also proves an analogous claim for the convex hull margin in Section 3.

**Lemma 2.** *For any $u \in \mathbb{R}^2$ let $d_u(x, y) = |\langle u, x - y\rangle|$. For any $\eta > 0$ there exists a clustering $\mathcal{C} = (C_1, C_2)$ on a set $X \subset \mathbb{R}^2$ that has arbitrarily large one-versus-all margin with respect to $d_{(0,1)}, d_{(1,0)}$, and yet $d_u(C_1, C_2) \le \eta\,\phi_{d_u}(X)$ for all $u \in \mathbb{R}^2$.*

*Proof sketch.* Consider Figure 1. The two points along the x axis belong to $C_1$, and the two points along the y axis belong to $C_2$. The pseudometric $d_1 = d_{(0,1)}$ measures the distance along the y axis, and the pseudometric $d_2 = d_{(1,0)}$ measures the distance along the x axis. Therefore $d_1(C_1, C_2) > 0 = \phi_{d_1}(C_1)$, and $d_2(C_1, C_2) > 0 = \phi_{d_2}(C_2)$. Hence, the one-versus-all margin of $\mathcal{C}$ with respect to $d_1, d_2$ is unbounded. Yet, for any $\eta > 0$ and any $u \in \mathbb{R}^2$ we can make $d_u(C_1, C_2) \le \eta\,\phi_{d_u}(X)$ by placing the endpoints of the clusters arbitrarily near the origin. $\quad\square$

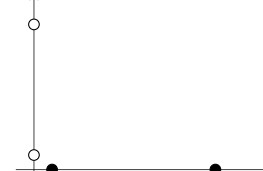

Figure 1: An instance with arbitrarily small SVM margin but unbounded one-versus-all margin.

## 4.1 The one-versus-all-margin captures the stability of center-based clusterings

Fix any pseudometric $d$ over $\mathcal{X}$. A clustering $\mathcal{C} = (C_1, \ldots, C_k)$ of $X$ is *center-based* if there exist $k$ points $c_1, \ldots, c_k \in \mathcal{X}$, called *centers*, such that for every $i \in [k]$ and every $x \in C_i$ we have $d(x, c_j) > d(x, c_i)$ for all $j \neq i$. In other terms, every point is assigned to the nearest center. It is well known that many popular center-based clustering problems, such as $k$-means or $k$-centers, are NP-hard to solve in general. However, those problems become polynomial-time solvable (or approximable) if the solution $\mathcal{C}$ meets certain stability properties. We show that two of these properties, the $\alpha$-center proximity of Awasthi et al. [2012] and the $(1 + \varepsilon)$-perturbation resilience of Bilu and Linial [2012], imply positive one-versus-all margin. Let us recall these properties. We define a $(1 + \varepsilon)$-perturbation of $d$ as any function $d'$ (which need not be a pseudometric) such that $d \leq d' \leq (1 + \varepsilon)d$.

**Definition 4.** *Let $\mathcal{C}$ be a center-based clustering.*

- *$\mathcal{C}$ satisfies the $\alpha$-center proximity property with $\alpha > 1$ if, for all $i \in [k]$, for all $x \in C_i$ and all $j \neq i$ we have $d(x, c_j) > \alpha \, d(x, c_i)$.*

- *$\mathcal{C}$ is $(1 + \varepsilon)$-perturbation resilient with $\varepsilon > 0$ if it is induced by the same centers $c_1, \ldots, c_k$ under any $(1 + \varepsilon)$-perturbation of $d$.*

It is known that $(1+\varepsilon)$-perturbation resilience implies $\alpha$-center proximity with $\alpha = 1+\varepsilon$, see [Awasthi et al., 2012]. Our result is:

**Theorem 2.** *If $\mathcal{C}$ satisfies $\alpha$-center proximity, then it has one-versus-all margin $\gamma \geq \frac{(\alpha-1)^2}{2(\alpha+1)}$. Hence, if $\mathcal{C}$ satisfies $(1 + \varepsilon)$-perturbation stability, then it has one-versus-all margin $\gamma \geq \frac{\varepsilon^2}{2(\varepsilon+2)}$.*

## 4.2 Cluster recovery with one-versus-all margin

We conclude by showing how a clustering with one-versus-all margin $\gamma > 0$ can be recovered with $\mathcal{O}(\log n)$ queries. Our algorithm, MREC, draws uniform random points from $X$ and then selects the smallest hypothesis consistent with the points sampled from each cluster (see Section 2). We will show that the VC-dimension of the class from which that hypothesis is taken can be bounded in terms of $\frac{1}{\gamma}$. This implies that, at every round, MREC learns a constant fraction of labels without mistakes.

We need some further notation. Let $d$ be any pseudometric over $\mathcal{X}$. For any $X \subset \mathcal{X}$ and any $r > 0$, let $\mathcal{M}(X, r, d)$ be the maximum cardinality of any $A \subseteq X$ such that $d(x, y) > r$ for all distinct $x, y \in A$. From now on we assume that $\mathcal{M}(X, r, d)$ is bounded[3], and for any $\gamma > 0$, we define $M(\gamma, d) = \max\{\mathcal{M}(B(x, r), \gamma r, d) : x \in \mathcal{X}, r > 0\}$. Thus any ball $B$ contains at most $M(\gamma, d)$ points at pairwise distance greater than $\gamma$ times the radius of $B$, and some $B$ attains this bound. Finally, by vc-dim$(H, X)$ we denote the VC-dimension of a generic concept class $H$ over a set $X$.

**Lemma 3** (One-versus-all margin implies one-sided-error learnability). *Let $d$ be any pseudometric over $\mathcal{X}$. For any finite $X \subset \mathcal{X}$ and any $\gamma > 0$, define the effective concept class over $X$:*

$$H = \{C \subseteq X \, : \, d(X \setminus C, C) > \gamma \, \phi_d(C)\} \tag{2}$$

*Then $H = I(H)$, and vc-dim$(H, X) \leq M^*(\gamma, d)$ where $M^*(\gamma, d) = \max(2, M(\gamma, d))$. Therefore, $H$ can be learned with one-sided error $\varepsilon$ and confidence $\delta$ with $\mathcal{O}\big(\varepsilon^{-2}(M^*(\gamma, d) \log 1/\varepsilon + 1/\delta)\big)$ examples by choosing the smallest consistent hypothesis in $H$.*

*Sketch of the proof.* To prove that $H = I(H)$, one can take any two $C_1, C_2 \in H$ and show that $C_1 \cap C_2$ satisfies the margin condition, too. To prove that vc-dim$(H, X) \leq M^*(\gamma, d)$, we have two steps. First, let sl$(H, X)$ be the *slicing dimension* of $H$. This is the size of the largest subset $S \subseteq X$ *sliced* by $H$, i.e., such that for every $x \in S$ there is $C \in H$ giving $S \setminus x = S \cap C$, see [Kivinen, 1995]. As the same work shows, we have vc-dim$(I(H), X) \leq$ sl$(H, X)$; hence, to prove vc-dim$(H, X) \leq M^*(\gamma, d)$ it suffices to prove that sl$(H, X) \leq M^*(\gamma, d)$. To this end, we use a packing argument. Suppose that $S \subseteq X$ is sliced by $H$, choose any $x \in S$, and let $C \in H$ such that $S \setminus x = S \cap C$. By construction of $H$, we know that $d(C, x) > \gamma \phi_d(C)$. Since $S \setminus x \subseteq C$, this yields:

$$d\big(S \setminus x, x\big) \geq d(C, x) > \gamma \, \phi_d(C) \geq \gamma \, \phi_d(S \setminus x) \tag{3}$$

---

[3]If $\mathcal{M}(X, r, d)$ is not bounded, then our results can be extended in the natural way, that is, we can prove a lower bound of $\Omega(n)$ queries for instances of $n$ points.

It can be shown that this implies $d(S \setminus x, x) > \gamma \phi_d(S)$ for all $x \in S$, and, in turn, $|S| \le M^*(\gamma, d)$. The claim on the learnability with one-sided error holds by choice of the smallest consistent hypothesis in $H = I(H)$, combined with standard PAC bounds. $\qquad\square$

We present our main result. Let $M(\gamma) = \max_{d \in \{d_1, \ldots, d_k\}} M(\gamma, d)$ with $d_1, \ldots, d_k$ as in Definition 3.

**Theorem 3.** *Let $(X, O)$ be any instance whose latent clustering $\mathcal{C}$ has one-versus-all margin $\gamma > 0$ with respect to $d_1, \ldots, d_k$. Then $\mathrm{MREC}(X, O, \gamma, d_1, \ldots, d_k)$ outputs $\mathcal{C}$ while making, with high probability, at most $\mathcal{O}(M^*(\gamma) k \log k \log n)$ label queries to $O$, where $M^*(\gamma) = \max(2, M(\gamma))$. Moreover, for any algorithm $\mathcal{A}$ and for any $\gamma > 0$, there are instances with one-versus-all margin $\gamma$ on which $\mathcal{A}$ makes $\Omega(M(2\gamma))$ label queries in expectation.*

*Sketch of the proof.* For the lower bounds, we take a set $X$ on $M(2\gamma)$ points at pairwise distance larger than $2\gamma$ times the radius of $X$, which is at least $\gamma$ times the diameter of $X$, and we draw a random clustering $\mathcal{C}$ in the form $(x, X \setminus x)$. One can see that $\mathcal{C}$ has one-versus-all margin $\gamma$, and simple arguments, coupled with Yao's principle for Monte Carlo algorithms, show that any algorithm needs $\Omega(M(2\gamma))$ queries in the worst case to return $\mathcal{C}$.

For the upper bounds, we show how to learn an expected constant fraction of $X$ with one-sided error using $\Theta(M^*(\gamma) k \log k)$ queries; the rest follows by our general boosting argument. To begin, for each $i \in [k]$ we consider the effective concept class:

$$H_i = \{C \subseteq X \,:\, d_i(C, X \setminus C) > \gamma \, \phi_{d_i}(C)\} \tag{4}$$

We then set $\varepsilon = 1/2k$ and $\delta = 1/2$, and draw a labeled sample $S$ of size $\Theta(\varepsilon^{-1}(M^*(\gamma) \log 1/\varepsilon + \log 1/\delta)) = \Theta(M^*(\gamma) k \log k)$. Finally, for each $i \in [k]$ we choose the smallest hypothesis $\widehat{C}_i \in H_i$ consistent with the subset $S_i \subseteq S$ labeled as $i$, and we assign label $i$ to all points in $\widehat{C}_i$. Clearly, $\widehat{C}_i \subseteq C_i$, therefore we are learning with one-sided error. Moreover, by Lemma 3, with probability at least $1/2$ we have $|\widehat{C}_i| \ge |C_i| - \varepsilon|X| = |C_i| - |X|/2k$. As $|\widehat{C}_i| \ge 0$, this implies $\mathbb{E}|\widehat{C}_i| \ge |C_i|/2 - |X|/4k$. By summing over all $i$, this shows that we are labelling correctly at least $|X|/4$ points in expectation. $\qquad\square$

**Remark.** By Theorem 3, in $\mathbb{R}^m$ MREC yields a $\mathcal{O}(\log n)$ query bound even when the clusters are *not* convex. Note however that this does not mean that MREC subsumes CHEATREC, for two reasons. First, as noted above, here $d_1, \ldots, d_k$ are known in advance, whereas MREC does not need to know them. Second, MREC works by computing the smallest hypothesis $\widehat{C}_i$ consistent with $S_i$ (see the proof of Theorem 3), which in general may take superpolynomial time. Indeed, CHEATREC runs in polynomial time by *not* computing $\widehat{C}_i$ at all.

## 5 One-versus-all clusterings

In this section we consider the case where the clusters of $\mathcal{C}$ are realized by some concept class $\mathcal{H}$ over $\mathcal{X}$. This is the clustering equivalent of the one-versus-all multiclass classifiers (see, e.g., [Shalev-Shwartz and Ben-David, 2014]). We show that $\mathcal{C}$ is actively recoverable if and only if $\mathcal{H}$ has finite *coslicing dimension*, a combinatorial quantity similar to the star number of Hanneke and Yang [2015] and the slicing dimension of Kivinen [1995]. We also show that, for a wide family of concept classes in $\mathbb{R}^m$, a finite coslicing dimension is equivalent to a positive one-versus-all margin.

Let $\mathcal{X}$ be any domain, $X \subset \mathcal{X}$ any finite set, and $\mathcal{C} = (C_1, \ldots, C_k)$ a clustering of $X$. Let $\mathcal{H}$ be any concept class over $\mathcal{X}$. We say that $\mathcal{C}$ is realized by $\mathcal{H}$ if for all $i \in [k]$ there is some $h_i \in \mathcal{H}$ such that $C_i = X \cap h_i$. For example, the ellipsoidal clusters of Bressan et al. [2020] can be formulated by letting $\mathcal{X} = \mathbb{R}^m$ and letting $\mathcal{H}$ to be the family of all ellipsoids in $\mathbb{R}^m$, and the convex clusters of Section 3 can be formulated by letting $\mathcal{X} = \mathbb{R}^m$ and letting $\mathcal{H}$ to be the family of all polytopes in $\mathbb{R}^m$. Clearly, we expect the number of queries needed to recover $\mathcal{C}$ to depend on $\mathcal{H}$. This leads us to the question: what can we say about the recoverability of $\mathcal{C}$ in terms of $\mathcal{H}$?

## 5.1 The coslicing dimension

We show that the number of queries needed to actively recover $\mathcal{C}$ depends on a combinatorial quantity that we call *coslicing dimension* of $\mathcal{H}$.

**Definition 5.** *We say that $\mathcal{H}$ coslices $X \subseteq \mathcal{X}$ if for all $x \in X$ there exist two concepts $h_x^+, h_x^- \in \mathcal{H}$ such that $X \cap h_x^+ = \{x\}$ and $X \cap h_x^- = X \setminus \{x\}$. The coslicing dimension of $\mathcal{H}$ is:*

$$\mathrm{cosl}(\mathcal{H}) = \sup\{|X| : X \text{ is cosliced by } \mathcal{H}\}$$

*If $\mathcal{H}$ coslices arbitrarily large sets then we let $\mathrm{cosl}(\mathcal{H}) = \infty$.*

For instance, let $\mathcal{X} = \mathbb{R}^m$. If $\mathcal{H}$ is the class of linear separators, then $\mathrm{cosl}(\mathcal{H}) = \infty$ (take $X$ to be the set of vertices of an $n$-vertex polytope and use the hyperplane separator theorem). If instead $\mathcal{H}$ is the class of axis-aligned boxes, it can be shown that $\mathrm{cosl}(\mathcal{H}) = 2m$. Our main result is:

**Theorem 4.** *If $\mathrm{cosl}(\mathcal{H}) < \infty$ then there is an algorithm that, given any $n$-point instance whose latent clustering $\mathcal{C}$ is realized by $\mathcal{H}$, recovers $\mathcal{C}$ with $\mathcal{O}(\mathrm{cosl}(\mathcal{H}) \, k \log k \log n)$ label queries with high probability. Moreover, for any algorithm $\mathcal{A}$ there are instances on $\mathrm{cosl}(\mathcal{H})$ points whose latent clustering $\mathcal{C}$ is realized by $\mathcal{H}$ where $\mathcal{A}$ makes $\Omega(\mathrm{cosl}(\mathcal{H}))$ label queries in expectation to return $\mathcal{C}$. As a consequence, if $\mathrm{cosl}(\mathcal{H}) = \infty$ then in the worst case any algorithm needs $\Omega(n)$ label queries in expectation to recover an $n$-point clustering realized by $\mathcal{H}$.*

*Sketch of the proof.* We follow the same ideas of the proof of Theorem 3. For the lower bound, we take any $X$ cosliced by $\mathcal{H}$ with $|X| = \mathrm{cosl}(\mathcal{H})$, and we draw a random clustering of $X$ in the form $(x, X \setminus x)$.

For the upper bound, for each $i \in [k]$ we define the effective concept class:

$$H_i = \{C \, : \, C = C_i' \wedge (C_1', \ldots, C_k') \in P_k(X)\} \tag{5}$$

where $P_k(X)$ is the set of all clusterings of $X$ realized by $\mathcal{H}$. As observed in the proof of Lemma 3, we have the general relationship $\mathrm{vc\text{-}dim}(I(H_i), X) \leq \mathrm{sl}(H_i, X)$ whenever $\mathrm{sl}(H_i, X) < \infty$. Therefore, if we show that $\mathrm{sl}(H_i, X) \leq \mathrm{cosl}(\mathcal{H})$, by drawing a labeled sample of size $\Theta(\mathrm{cosl}(\mathcal{H}) \, k \log k)$ we can recover the labels of an expected constant fraction of $X$, as in the proof of Lemma 3. To prove that $\mathrm{sl}(H_i, X) \leq \mathrm{cosl}(\mathcal{H})$, let $U = \{x_1, \ldots, x_\ell\} \subseteq X$ be sliced by $H_i$. By construction of $H_i$, there are $\ell$ clusterings $\mathcal{C}_1, \ldots, \mathcal{C}_\ell$ realized by $\mathcal{H}$ and such that $\mathcal{C}_j = (x_j, U \setminus x_j)$ for all $j \in [\ell]$. This implies that $U$ is cosliced by $\mathcal{H}$. Hence, $|U| \leq \mathrm{cosl}(\mathcal{H})$ and so $\mathrm{sl}(H_i, X) \leq \mathrm{cosl}(\mathcal{H})$. $\qquad\square$

**Remark.** In the case of convex clusters in $\mathbb{R}^m$, we would let $\mathcal{H}$ be the class of all convex polytopes, obtaining $\mathrm{cosl}(\mathcal{H}) = \infty$ and thus Theorem 4 would not provide any useful bound. This is true even if $\mathcal{C}$ has convex hull margin $\gamma > 0$, although by Theorem 1 we know that $\mathcal{C}$ can be recovered with $\mathcal{O}(\log n)$ queries. The same holds for the one-versus-all margin. This is because we defined the coslicing dimension as a function of $\mathcal{H}$, through which we cannot capture margin properties — indeed, the margin depends on the instance $(X, O)$ rather than on $\mathcal{H}$. We note however that one could fix this by redefining the coslicing dimension in the form $\mathrm{cosl}(\mathcal{H}, \mathcal{I})$, where $\mathcal{I}$ is a class of instances, and adapt Theorem 4 correspondingly. Then, under the assumption that all the instances $(X, O) \in \mathcal{I}$ have margin $\gamma > 0$, one could bound $\mathrm{cosl}(\mathcal{H}, \mathcal{I})$ as a function of $\gamma$, recovering the same type of bounds of Theorem 1 and Theorem 3.

## 5.2 The one-versus-all margin, again!

We look again at the case $\mathcal{X} = \mathbb{R}^m$. Consider any concept class $\mathcal{H}$. Theorem 4 and the remark above say that, if $\mathrm{cosl}(\mathcal{H}, \mathcal{I}) < \infty$ where $\mathcal{I}$ is the class of allowed instances, then the latent clustering of any instance can be recovered with $\mathcal{O}(\log n)$ queries. Note that we are not using any notion of margin here — only the finiteness of $\mathrm{cosl}(\mathcal{H}, \mathcal{I})$.

Here we show that, for a wide family of concept classes in $\mathbb{R}^m$, a finite coslicing dimension and a positive one-versus-all margin are actually equivalent. This equivalence is established by proving that the $\mathcal{O}(\log n)$ query bound is attainable *if and only* if the instances have positive one-versus-all margin. In what follows, we assume that $\mathcal{H}$ satisfies:

**Definition 6.** *A concept class $\mathcal{H}$ in $\mathbb{R}^m$ is non-fractal if there is $h \in \mathcal{H}$ such that both $h$ and its complement contain a ball of positive radius.*

This assumption avoids pathological cases; for instance, $\mathcal{H}$ containing only hypotheses that are affine transformations of Cantor's set. Our result is:

**Theorem 5.** *Let $\mathcal{H}$ be a concept class in $\mathbb{R}^m$ that is non-fractal and closed under affine transformations. There is an algorithm that, given any instance whose latent clustering $\mathcal{C}$ has one-versus-all margin $\gamma$ and is realized by $\mathcal{H}$, returns $\mathcal{C}$ while making $\mathcal{O}(Mk \log k \log n)$ label queries with high probability, where $M = \max\big(2, (1 + 4/\gamma)^m\big)$. Moreover, for any algorithm $\mathcal{A}$, there exist arbitrarily large $n$-point instances, whose latent clustering $\mathcal{C}$ has arbitrarily small one-versus-all margin and is realized by $\mathcal{H}$, where $\mathcal{A}$ makes $\Omega(n)$ label queries in expectation to recover $\mathcal{C}$.*

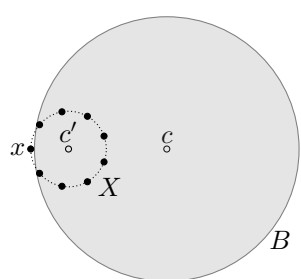

Figure 2: The $\eta$-packing $X$ is in $B$, and thus in $h$, except for $x$ that lies in $\overline{h}$. By taking $B$ arbitrarily close to $x$, we can make $\eta$ arbitrarily small and thus $X$ arbitrarily large.

*Sketch of the proof.* The upper bounds follow by Theorem 3 and the packing number of $\mathbb{R}^m$. For the lower bounds, we show that arbitrarily large packings of a sphere are cosliced by $\mathcal{H}$. We use Figure 2 for reference. Let $h \in \mathbb{R}^m$ be such that both $h$ and its complement $\overline{h}$ contain a ball of positive radius. Then, for any $\rho > 0$ there exist a ball $B = B(c, r) \subseteq h$ with $r > 0$, and a point $x \in \overline{h}$ such that $d(B, x) \le \rho$. Now take a sphere $S$ of radius $r' \ll r$ with center on the segment $xc$. Let $\eta = \sup_{y \in S \setminus B} d(x, y)$, and let $X$ be an $\eta$-packing of $S$, that is, a subset of points of $S$ such that $d(x', x'') > \eta$ for all distinct $x', x'' \in X$. Note how this implies that every $x' \in X \setminus x$ necessarily lies in $S \cap B$, and therefore, in $S \cap h$. Moreover, by letting $\eta/r' \to 0$, we can take $X$ arbitrarily large. Since $\mathcal{H}$ is closed under affine transformations, by rotating $X$ it follows that for every $x \in X$ there is $h_x \in \mathcal{H}$ such that $X \cap h_x = X \setminus x$. By applying the same argument to $\overline{h}$ and by complementation we can show that $X \cap h'_x = \{x\}$ for some $h'_x \in \mathcal{H}$ for all $x \in X$ as well. Hence $\mathcal{H}$ coslices arbitrarily large sets. To conclude, invoke Theorem 4. $\square$

Note that Theorem 5 applies to several basic concept classes $\mathcal{H}$. For instance, when $\mathcal{H}$ is the class of all linear separators, the class of all ellipsoids, the class of all polytopes, and the class of all convex bodies (bounded or not, and possibly degenerate). It also includes more complex classes with non-convex concepts, for instance, the class of all finite or infinite disjoint unions of balls, polytopes, or convex bodies.

# 6 Future Work

An interesting direction for future work is to explore the power of different types of queries to avoid the exponential dependency on the dimensionality of the ambient space. Another open problem is to understand if our margin conditions are sufficient to obtain PTASes for k-means and other center-based clustering problems. A third open problem is to close the gap in the exponential dependence on the dimension in the Euclidean case, which is $(1+1/\gamma)^m$ in our upper bound and roughly $(1+1/\gamma)^{m/2}$ in the lower bound of Bressan et al. [2020].

## Acknowledgments and Disclosure of Funding

The authors gratefully acknowledge partial support by the Google Focused Award "Algorithms and Learning for AI" (ALL4AI). Nicolò Cesa-Bianchi is also supported by the MIUR PRIN grant Algorithms, Games, and Digital Markets (ALGADIMAR) and by the EU Horizon 2020 ICT-48 research and innovation action under grant agreement 951847, project ELISE (European Learning and Intelligent Systems Excellence).

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
