# On Margin-Based Cluster Recovery
# with Oracle Queries
# (Supplementary Material)

**Marco Bressan**
Dept. of CS, Univ. of Milan, Italy
marco.bressan@unimi.it

**Nicolò Cesa-Bianchi**
DSRC & Dept. of CS, Univ. of Milan, Italy
nicolo.cesa-bianchi@unimi.it

**Silvio Lattanzi**
Google
silviol@google.com

**Andrea Paudice**
Dept. of CS, Univ. of Milan, Italy &
Istituto Italiano di Tecnologia, Italy
andrea.paudice@unimi.it

## 1   Proof of Lemma 1

**Lemma 1** (Convex hull expansion trick). *Fix $\gamma > 0$, and let $s = \Theta\big(m^5\big(1 + 1/\gamma\big)^m \log\big(1 + 1/\gamma\big)\big)$ large enough. Let $S_C$ be a sample of $s$ independent uniform random points from some cluster $C$, and let $K = \mathrm{conv}(S_C)$. Let $X_1, \ldots, X_N$ be independent random points sampled $\varepsilon$-uniformly from $K$, with $\varepsilon \in \Theta(m^{-1})$ small enough and $N \in \Theta(m^2)$ large enough, and let $z = \frac{1}{N}\sum_{i=1}^N X_i$. Finally, let $Q = (1 + \gamma)K$ where the center of the scaling is $z$. Then $\mathbb{P}\big(|Q \cap C| \geq |C|/2\big) \geq 1/2$.*

**Preliminaries.** Without loss of generality, we assume that $K$ has full rank (as one can always work in the subspace spanned by $S_C$, which can be computed in time $\mathcal{O}(|S_C|m)$). For technical reasons, we let $\vartheta = \frac{1}{1+\gamma} \in (0, 1)$ and prove the lemma for $s = \Omega\big(\frac{m^5}{\vartheta(1-\vartheta)^m} \ln \frac{1}{\vartheta(1-\vartheta)}\big)$ large enough. To see that any $s \in \Theta\big(m^5\big(1 + 1/\gamma\big)^m \ln\big(1 + 1/\gamma\big)\big)$ satisfies this assumption, first substitute $\vartheta$ to get:

$$\frac{1}{\vartheta(1-\vartheta)^m} \ln \frac{1}{\vartheta(1-\vartheta)} = (1+\gamma)\left(\frac{1+\gamma}{\gamma}\right)^m \ln \frac{(1+\gamma)^2}{\gamma} \tag{6}$$

which is in $\mathcal{O}\big((1 + 1/\gamma)^m(1 + \gamma)\ln(1 + 1/\gamma)\big)$. Now note that $(1 + \gamma)\ln\big(1 + 1/\gamma\big)$ is bounded by $\mathcal{O}\big(\gamma \cdot 1/\gamma\big) = \mathcal{O}(1)$ for $\gamma > 1$, and by $2\ln\big(1 + 1/\gamma\big) = \mathcal{O}\big(\ln 1/\gamma\big)$ when $\gamma \leq 1$. Hence, in any case the term $(1 + \gamma)\ln(1 + 1/\gamma)$ is in $\mathcal{O}(\ln(1 + 1/\gamma))$. Therefore:

$$\big(1 + 1/\gamma\big)^m \ln\big(1 + 1/\gamma\big) = \Omega\left(\frac{1}{\vartheta(1-\vartheta)^m} \ln \frac{1}{\vartheta(1-\vartheta)}\right) \tag{7}$$

as claimed.

Before starting with the actual proof, we introduce some further definitions and notation.

**Definition 7.** *A convex body $K \subset \mathbb{R}^m$ is in* isotropic position[4] *if it has center of mass in the origin, $\int_K x \, dx = 0$, and moment of inertia 1 in every direction, $\int_K \langle x, u \rangle^2 dx = 1$ for all $u \in S^{m-1}$.*

We define the norm $\|\cdot\|_K = \|f(\cdot)\|_2$ where $f = f_K$ is the unique affine transformation such that $f(K)$ is in isotropic position. We let $R(K) = \sup_{x \in K} \|x\|_2$ denote the Euclidean radius of $K$, and we let $R_K(K) = \sup_{x \in K} \|x\|_K$ denote the isotropic radius of $K$. We also let $d_K(x, y) = \|x - y\|_K$ be the isotropic distance of $K$.

Now, the proof has two steps. First, we show that the point $z = \frac{1}{N}\sum_{i=1}^N X_i$ is close to the centroid $\mu_K$ of $K$, according to $d_K(\cdot)$, with good probability. Second, we show that if this is the case, then

---

[4]Not to be confused with the definition of Giannopoulos [2003], where the assumption $\int_K \langle x, u \rangle^2 dx = 1$ is replaced by $\mathrm{vol}(K) = 1$.

$\frac{K}{\vartheta}$, where the scaling is meant about $z$, contains a polytope $P$ which contains $K$ and thus $S_C$, and which belongs to a class with VC dimension $\mathcal{O}\big(\frac{m^5}{\vartheta(1-\vartheta)^m}\big)$. By standard PAC bounds this implies that $|P \cap C| \geq \frac{1}{2}|C|$, with a probability that can be made arbitrarily close to 1 by adjusting the constants.

**Step 1: $z$ is close to the centroid of $K$**

Let $\mu_K$ be the center of mass of $K$. We prove:

**Lemma 4.** *Fix any $\eta, p \geq 0$, and choose any $\varepsilon \leq \frac{\eta}{4(m+1)}$ and $N \geq \frac{16(m+1)^2}{p^2\eta^2}$. If $X_1, \ldots, X_N$ are drawn independently and $\varepsilon$-uniformly at random from $K$, and $\overline{X} = \frac{1}{N}X_i$, then:*

$$\mathbb{P}\big(d_K\big(\overline{X}, \mu_K\big) \leq \eta\big) \geq 1 - p \tag{8}$$

For the proof of Lemma 4, we need two ancillary results.

**Lemma 5.** $R_K(K) \leq m + 1$.

*Proof.* Consider $K$ in isotropic position, and let $K' = \vartheta K$ where $\vartheta = \mathrm{vol}(K)^{-1/m}$, so that $\mathrm{vol}(K') = 1$. Then, $K'$ is in isotropic position according to the definition of Giannopoulos [2003]. In this case, [Giannopoulos, 2003, Theorem 1.2.4] implies $R(K') \leq (m+1)L_K$, where $L_K$ is the *isotropic constant* which, for all $u \in S^{m-1}$, satisfies $\int_{K'} \langle x, u \rangle^2 \, dx = L_K^2$. Since $K' = \vartheta K$ and $\int_K \langle x, u \rangle^2 \, dx = 1$ by the isotropy of $K$, we have $L_K = \vartheta$. Hence $R(K') \leq (m+1)\vartheta$, that is, $R(K) \leq m + 1$. $\qquad\square$

**Lemma 6.** *If $X$ is drawn from an $\varepsilon$-uniform distribution over $K$, then $\|\mathbb{E}X\|_K \leq 2\varepsilon(m+1)$.*

*Proof.* Since $X$ is $\varepsilon$-uniform over $K$, there exists a coupling $(X, Y)$ with $\mathbb{P}(X \neq Y) \leq \varepsilon$ and $Y$ uniform over $K$. Since $\|\mathbb{E}Y\|_K = 0$, we have:

$$\|\mathbb{E}X\|_K = \|\mathbb{E}[X - Y]\|_K \leq \mathbb{P}(X \neq Y) \sup_{x,y \in K} d_K(x, y) \leq \varepsilon\, 2R_K(K) \leq 2\varepsilon(m+1) \tag{9}$$

where the last inequality is given by Lemma 5. $\qquad\square$

*Proof of Lemma 4.* For the sake of the analysis we look at $K$ from its isotropic position. Note that the $X_i$ are still $\varepsilon$-uniform over $K$, since the affine map that places $K$ in isotropic position preserves volume ratios. The claim becomes:

$$\mathbb{P}\big(\|\overline{X}\|_2 \leq \eta\big) \geq 1 - p \tag{10}$$

Now, $\|\overline{X}\|_2 \leq \|\mathbb{E}\overline{X}\|_2 + \|\overline{X} - \mathbb{E}\overline{X}\|_2$. Thus, we show that $\|\mathbb{E}\overline{X}\|_2 \leq \frac{\eta}{2}$, and that $\|\overline{X} - \mathbb{E}\overline{X}\|_2 \leq \frac{\eta}{2}$ with probability at least $1 - p$. For the first part, by Lemma 6, and since $\varepsilon \leq \frac{\eta}{4(m+1)}$, we obtain:

$$\|\mathbb{E}\overline{X}\|_2 = \|\mathbb{E}X_i\|_2 \leq 2\varepsilon(m+1) \leq \frac{\eta}{2} \tag{11}$$

For the second part, by Lemma 5 we have $\|X_i\|_2 \leq m+1$ for all $i$. Therefore, if we let $Y_i = X_i - \mathbb{E}X_i$ for all $i$, we have $\|Y_i\|_2 \leq 2(m+1)$ and thus $\|Y_i\|_2^2 \leq 4(m+1)^2$. Now let $\overline{Y} = \frac{1}{N}\sum_{i=1}^N Y_i$. Since the $Y_i$ are independent and with $\mathbb{E}Y_i = 0$, then $\mathbb{E}\langle Y_i, Y_j \rangle = 0$ whenever $i \neq j$ and therefore:

$$\mathbb{E}\|\overline{Y}\|_2^2 = \mathbb{E}\bigg(\frac{1}{N^2}\sum_{i,j=1}^N \langle Y_i, Y_j \rangle\bigg) = \frac{1}{N^2}\sum_{i=1}^N \mathbb{E}\|Y_i\|_2^2 \leq \frac{4(m+1)^2}{N} \tag{12}$$

Plugging in our value $N \geq \frac{16(m+1)^2}{p^2\eta^2}$, and using Jensen's inequality, we obtain:

$$\big(\mathbb{E}\|\overline{Y}\|_2\big)^2 \leq \mathbb{E}\|\overline{Y}\|_2^2 \leq \frac{p^2\eta^2}{4} \tag{13}$$

Therefore $\mathbb{E}\|\overline{Y}\|_2 \leq \frac{p\eta}{2}$, which by Markov's inequality implies that $\mathbb{P}\big(\|\overline{Y}\|_2 > \frac{\eta}{2}\big) < p$. By noting that $\overline{Y} = \overline{X} - \mathbb{E}\overline{X}$, the proof is complete. $\qquad\square$

**Step 2: showing a tight enclosing polytope**

We prove:

**Lemma 7.** *Let $z \in \mathbb{R}^m$ such that $d_K(z, \mu_K) \leq \frac{1}{e} - \frac{1}{3}$. For any $\vartheta \in (0,1)$ there exists a polytope $P$ on $t = \mathcal{O}\left(\frac{m^2}{\vartheta(1-\vartheta)^m}\right)$ vertices such that $K \subseteq P \subseteq \frac{K}{\vartheta}$, where the scaling is about $z$.*

For the proof, we need some ancillary results.

**Theorem 6** (Bertsimas and Vempala [2004], Theorem 3)**.** *Let $K$ be a convex set in isotropic position and $z$ be a point at distance $t$ from its centroid. Then any halfspace containing $z$ contains at least $\frac{1}{e} - t$ of the volume of $K$.*

Now we adapt a result by Naszódi [2018], recalled here for convenience. We say that a halfspace $F$ supports a convex body from outside if $F$ intersects the boundary of the body, but not its interior.

**Lemma 8** (Lemma 2.1, Naszódi [2018])**.** *Let $0 < \vartheta < 1$, and $F$ be a halfspace that supports $\vartheta K$ from outside, where the scaling is about $\mu_K$. Then:*

$$\mathrm{vol}(K \cap F) \geq \mathrm{vol}(K) \cdot (1 - \vartheta)^m \frac{1}{e} \tag{14}$$

Our adaptation is:

**Lemma 9.** *Let $z \in \mathbb{R}^m$, let $0 < \vartheta < 1$, and let $F$ be a half-space that supports $\vartheta K$ from outside, where the scaling is about $z$. Then:*

$$\mathrm{vol}(K \cap F) \geq \mathrm{vol}(K) \cdot (1 - \vartheta)^m \left( \frac{1}{e} - d_K(\mu_K, z) \right) \tag{15}$$

*Proof.* The proof is similar to the proof of the original lemma. See Figure 3 for reference. Let $F_0$ be a translate of $F$ whose boundary contains $z$, and let $K_0 = K \cap F_0$. Let $F_1$ be a translate of $F$ that supports $K$ from outside, and let $p \in F_1 \cap K$. Now consider $K_0' = \vartheta p + (1 - \vartheta) K_0$, that is, the homothetic copy of $K_0$ with center $p$ and ratio $1 - \vartheta$. The crucial observation is that $F = \vartheta p + (1 - \vartheta) F_0$, which implies $K_0' \subset K \cap F$. Clearly $\mathrm{vol}(K_0') = (1 - \vartheta)^m \mathrm{vol}(K_0)$. Moreover, by Theorem 6 we have $\mathrm{vol}(K_0) \geq \mathrm{vol}(K)(1/e - t)$ where $t = d_K(\mu_K, z)$; this holds because mapping $K$ to its isotropic position preserves volume ratios. This concludes the proof. $\square$

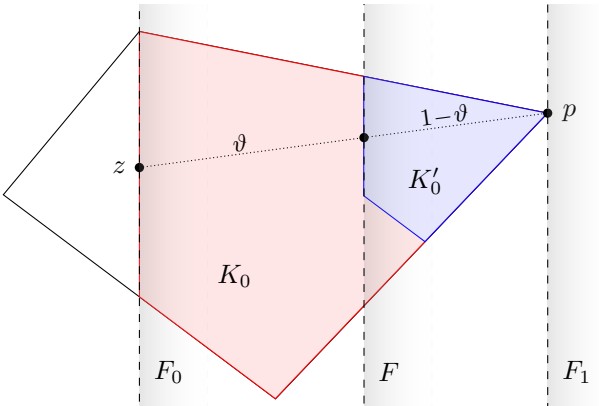

Figure 3: a visual proof of Lemma 9, with $d(z,p) = 1$ for simplicity.

*Proof of Lemma 7.* We adapt the construction behind [Naszódi, 2018, Theorem 1.2], by replacing $\varepsilon = \frac{(1-\vartheta)^m}{e}$ with $\varepsilon = \frac{(1-\vartheta)^m}{3}$. The theorem then says that, if we set:

$$t = \left\lceil C \frac{(m+1)3}{(1-\vartheta)^m} \ln \frac{3}{(1-\vartheta)^m} \right\rceil \tag{16}$$

and we let $X_1, \ldots, X_t$ be $t$ points chosen independently and uniformly at random from $K$, and let $P = \text{conv}(X_1, \ldots, X_t)$, then $\vartheta K \subseteq P \subseteq K$ with probability at least $1 - p$, where

$$p = 4 \left( 11 C^2 \left( \frac{(1-\vartheta)^m}{3} \right)^{C-2} \right)^{m+1} \tag{17}$$

Now, we choose $C = \Theta(\frac{1}{\vartheta} \ln \frac{1}{\vartheta})$ large enough. On the one hand, we obtain:

$$t = \left\lceil C \frac{(m+1)3}{(1-\vartheta)^m} \ln \frac{3}{(1-\vartheta)^m} \right\rceil = \mathcal{O}\left( \frac{m^2}{\vartheta(1-\vartheta)^m} \ln \frac{1}{\vartheta} \ln \frac{1}{1-\vartheta} \right) \tag{18}$$

Since $\vartheta \in (0,1)$, we have $\ln \frac{1}{\vartheta} \ln \frac{1}{1-\vartheta} = \ln(1 + 1/x) \ln(1 + x)$ where $x = \frac{\vartheta}{1-\vartheta} > 0$. However, $\ln(1 + 1/x) \ln(1 + x) < 1$ for all $x > 0$. Therefore, $t \in \mathcal{O}\left( \frac{m^2}{\vartheta(1-\vartheta)^m} \right)$. On the other hand, by setting $C$ large enough we can make $C^2 \left( \frac{(1-\vartheta)^m}{3} \right)^{C-2}$ arbitrarily small, and therefore $p < 1$.

Since $p < 1$, we conclude that *there exists* a polytope $P$ on $t \in \mathcal{O}\left( \frac{m^2}{\vartheta(1-\vartheta)^m} \right)$ vertices such that $\vartheta K \subseteq P \subseteq K$. To conclude, instead of $P$ simply take $\frac{P}{\vartheta}$ where the scaling is about $z$. $\qquad\square$

**Wrap-up**

First, by Lemma 4, by taking $N \in \mathcal{O}(m^2)$ large enough we can make $d_K(\mu_K, z) \leq \frac{1}{e} - \frac{1}{3}$ with probability arbitrarily close to 1. Now let $\mathcal{P}_t$ be the family of all polytopes in $\mathbb{R}^m$ on at most $t$ vertices. For $t \in \mathcal{O}\left( \frac{m^2}{\vartheta(1-\vartheta)^m} \right)$ large enough, Lemma 7 implies that there exists some $P \in \mathcal{P}_t$ such that $K \subseteq P \subseteq \frac{K}{\vartheta}$.

Now we prove that, by choosing $s$ large enough, with probability arbitrarily close to 1 we have $\left| \frac{K}{\vartheta} \cap C \right| \geq \frac{1}{2}|C|$. First, by a recent result of Kupavskii [2020], we have $\text{vc-dim}(\mathcal{P}_t) \leq 8m^2 t \log_2 t$. For our $t$ this yields

$$\text{vc-dim}(\mathcal{P}_t) = \mathcal{O}\left( m^2 \frac{m^2}{\vartheta(1-\vartheta)^m} \ln \frac{m^2}{\vartheta(1-\vartheta)^m} \right) = \mathcal{O}\left( \frac{m^5}{\vartheta(1-\vartheta)^m} \ln \frac{1}{\vartheta(1-\vartheta)} \right) \tag{19}$$

where in the last equality we used $\ln \frac{m^2}{\vartheta(1-\vartheta)^m} = \ln \frac{m^{(2/m)m}}{\vartheta(1-\vartheta)^m} \leq m \ln \frac{m^{2/m}}{\vartheta(1-\vartheta)} = \mathcal{O}\left( m \ln \frac{1}{\vartheta(1-\vartheta)} \right)$. Hence, by choosing $|S_C| = s = \mathcal{O}\left( \frac{m^5}{\vartheta(1-\vartheta)^m} \ln \frac{1}{\vartheta(1-\vartheta)} \right)$ large enough, for any constant $c, \varepsilon, \delta$ we can make:

$$|S_C| \geq c \frac{\text{vc-dim}(\mathcal{P}_t) \ln \frac{1}{\varepsilon} + \ln \frac{1}{\delta}}{\varepsilon} \tag{20}$$

Since $P$ is consistent with $S_C$, that is, $P \supset S_C$, then by standard PAC bounds we have $|P \cap C| \geq (1 - \varepsilon)|C|$ with probability at least $1 - \delta$. But $P \subseteq \frac{K}{\vartheta}$, and therefore $|\frac{K}{\vartheta} \cap C| \geq (1 - \varepsilon)|C|$ with probability at least $1 - \delta$. By adjusting the constants this yields the thesis of Lemma 1.

## 2 Proof of Theorem 1

**Theorem 1.** *Let $(X, O)$ be an instance whose latent clustering $\mathcal{C}$ has convex hull margin $\gamma > 0$. Then* CHEATREC$(X, O, \gamma)$ *outputs $\mathcal{C}$, runs in time* $\text{poly}(k, n, m)$, *and with high probability makes a number of label queries to $O$ bounded by* $\mathcal{O}\left( k^2 m^5 \left( 1 + 1/\gamma \right)^m \log(1 + 1/\gamma) \log n \right)$.

We give the pseudocode of the algorithm for reference. First, we prove the correctness and the query bound. Then we show the running time bound. Note that, for readability, the pseudocode given here is high-level; the actual implementation is more complex, see below.

**Correctness and query bound.** We prove that, at each round, for some $i$ we recover at least half of the points in $C_i$ with probability $1 - \delta$, where $\delta$ can be made arbitrarily small by adjusting the constants. Let $S_i$ be the subset of the sample $S$ having label $i$. Since there are at most $k$ clusters and $|S| = ks$, for some $i$ we will have $|S_i| \geq s$. Now we apply Lemma 1 to $K = \text{conv}(S_i)$. Since $s$ satisfies the hypotheses, the lemma says that $\widehat{C}_i = Q \cap X$ has size $|\widehat{C}_i| \geq \frac{|C_i|}{2}$ with probability

---

**Algorithm 1** HULLTRICK$(K, 1 + \gamma)$

---

let $N = \Omega(m^2)$ large enough
let $\varepsilon = \mathcal{O}(m^{-1})$ small enough
draw $N$ i.i.d. random points $X_1, \dots, X_N$ from any $\varepsilon$-uniform distribution over $K$
let $z = \frac{1}{N} \sum_{i=1}^{N} X_i$
**return** $z + (1 + \gamma)(K - z)$

---

**Algorithm 2** CHEATREC$(X, O, \gamma)$

---

**while** $X \neq \emptyset$ **do**
    let $s = \mathcal{O}\big(m^5 \big(1 + \frac{1}{\gamma}\big)^m \ln \big(1 + \frac{1}{\gamma}\big)\big)$ large enough
    draw a uniform random sample $S$ of size $\min(|X|, ks)$ from $X$, without repetition
    learn the labels of $S$ with $ks$ queries to $O$
    let $S_i$ be the points of $S$ having label $i$
    **for** $i = 1, \dots, k$ **do**
        $K = \mathrm{conv}(S_i)$
        $Q = $ HULLTRICK$(K, 1 + \gamma)$
        $\widehat{C}_i = Q \cap X$
        label all points of $\widehat{C}_i$ with label $i$
        $X = X \setminus \widehat{C}_i$

---

arbitrarily close to 1 (that is, with probability $1 - \delta$ as above). It remains to show that $\widehat{C}_i \subseteq C_i$. Let $d$ be any pseudometric that is homogeneous and invariant under translation. Then, any point $y \in (1 + \gamma)K \cap X$ satisfies $d(y, K) \leq \gamma \phi_d(K)$. But $K = \mathrm{conv}(S_i) \subseteq \mathrm{conv}(C_i)$. Therefore $\phi_d(K) \leq \phi_d(C_i)$. Hence $d(y, \mathrm{conv}(C_i)) \leq \gamma \phi_d(C_i)$. By the convex margin assumption, this implies that $y \in C_i$. This also proves the correctness of the algorithm. The total query bound follows as in Lemma 3 of [Bressan et al., 2020], whose algorithm also recovers an expected fraction $\frac{1}{4k}$ of all points in each round.

**Running time bound.** First, we analyze the running time of CHEATREC excluding the call to HULLTRICK. Drawing and labeling the samples obviously cost $\mathcal{O}(n)$ time throughout the entire execution. In the main loop, $K$ is actually not computed explicitly — see below. Similarly, $Q$ is simply a set of points obtained by rescaling $S_i$ about some point in space. Thus, computing $\widehat{C}_i = Q \cap X$ and labeling its points boils down to deciding, for all $x \in X$, if $x$ can be written as $\sum_{x_j \in Q_j} \lambda_j x_j$ for a set of coefficients $\lambda_1, \dots, \lambda_{|Q|} \in [0, 1]$. This can be done with polynomial precision using any polynomial-time solver for linear programs (say, the ellipsoid method).

Let us now turn to HULLTRICK. The computationally expensive part is drawing $N$ points from an $\varepsilon$-uniform distribution over $K = \mathrm{conv}(S_i)$. This can be done with any method of choice. Here, we consider the "hit-and-run from a corner" algorithm of Lovász and Vempala [2006], which implements a fast mixing random walk whose stationary distribution is uniform over any convex body. We remark that other methods for computing approximate centers exist, see for example [Basu and Oertel, 2017].

The implementation is as follows. First, we put $K$ in near-isotropic position by computing the minimum volume enclosing ellipsoid (MVEE) and then applying an affine transformation to make the MVEE into the ball of unit radius. As shown in [Khachiyan, 1996], this operation takes time $|S_i| m^2 \big(\ln m + \ln \ln |S_i|\big)$. After this transformation, let $\mu$ be the center of the unit ball that contains $K$. Observe that $\mu$ is at distance at least $\frac{1}{m}$ from the boundary of $K$: this holds since, by John's theorem, the ball of radius $\frac{1}{m}$ centered at $\mu$ is entirely contained in $K$. Now we execute the hit-and-run from a corner algorithm starting at $\mu$. By the results of Lovász and Vempala [2006], we have the following bound.

**Lemma 10** (See Lovász and Vempala [2006], Corollary 1.2). *Assume hit-and-run is started from $\mu$. For any $\varepsilon > 0$, the distribution of the random walk after*

$$t = \Theta\left(m^5 \ln \frac{m}{\varepsilon}\right)$$

*steps is $\varepsilon$-uniform over $K$.*

It remains to implement the hit-and-run algorithm over $K$. To this end we need to solve the following problem: given a generic point $x \in K$ and a vector $u \in S^{n-1}$, determine the intersection of the ray $\{x + \alpha u\}_{\alpha \geq 0}$ with the boundary of $K$. This amounts to solving a linear program that searches for the maximum value $\alpha \geq 0$ such that $x + \alpha u$ can be written as $\sum_{x_j \in S_i} \lambda_j x_j$ for a set of coefficients $\lambda_1, \ldots, \lambda_{|S_i|} \in [0, 1]$. We can solve such an LP in time $t_K = \text{poly}(|S_i|, m)$ with polynomial precision using any polynomial-time solver for linear programs. The total time to draw the $N$ samples is therefore:

$$\mathcal{O}\left( |S_i| m^2 \left( \ln m + \ln \ln |S_i| \right) + N t_K \, m^5 \ln \frac{m}{\varepsilon} \right) \tag{21}$$

As we set $N = \mathcal{O}(m^2)$ and $\varepsilon = \mathcal{O}(m^{-1})$, the total running time of HULLTRICK is:

$$\mathcal{O}\left( |S_i| m^2 \left( \ln m + \ln \ln |S_i| \right) + t_K \, m^7 \ln m \right) \tag{22}$$

which is in $\text{poly}(|S_i|, m) = \text{poly}(n, m)$.

## 3 Proof of Lemma 2

**Lemma 2.** *For any $u \in \mathbb{R}^2$ let $d_u(x, y) = |\langle u, x - y \rangle|$. For any $\eta > 0$ there exists a clustering $\mathcal{C} = (C_1, C_2)$ on a set $X \subset \mathbb{R}^2$ that has arbitrarily large one-versus-all margin with respect to $d_{(0,1)}, d_{(1,0)}$, and yet $d_u(C_1, C_2) \leq \eta \, \phi_{d_u}(X)$ for all $u \in \mathbb{R}^2$.*

Let $u_1 = (1, 0)$ and $u_2 = (0, 1)$, and for some constant $a$ independent of $\eta$ and to be fixed later, consider the set $X$ consisting of the four points (see Figure 1):

$$p_1 = \eta \, u_1, \quad q_1 = a \, u_1 \tag{23}$$
$$p_2 = \eta \, u_2, \quad q_2 = a \, u_2 \tag{24}$$

Finally, let $\mathcal{C} = (C_1, C_2)$ where $C_1 = \{p_1, q_1\}$ and $C_2 = \{p_2, q_2\}$.

Consider the two pseudometrics $d_1 = d_{(0,1)}$ and $d_2 = d_{(1,0)}$. Then $\phi_{d_1}(C_1) = 0$ and $d_1(C_1, C_2) = \eta$, and vice versa, $\phi_{d_2}(C_2) = 0$ and $d_2(C_1, C_2) = \eta$. Thus, the one-versus-all margin of $\mathcal{C}$ with respect to $d_1, d_2$ is unbounded.

Now choose any $u \in \mathbb{R}^2 \setminus \mathbf{0}$. Without loss of generality, by rescaling we can assume $u$ to be a unit vector. In this case, we have $d_u(C_1, C_2) \leq d_u(p_1, p_2) \leq \|p_1 - p_2\|_2 = \eta \sqrt{2}$. Yet, the convex hull of $X$ contains a ball of radius $\Omega(1)$, and therefore $\phi_d(X) = \Omega(1)$, where the constants depend on $a$. Hence, we can make $d(C_1, C_2) \leq \eta \, \phi_d(X)$ by choosing $a$ appropriately.

## 4 Proof of Theorem 2

**Theorem 2.** *If $\mathcal{C}$ satisfies $\alpha$-center proximity, then it has one-versus-all margin $\gamma \geq \frac{(\alpha-1)^2}{2(\alpha+1)}$. Hence, if $\mathcal{C}$ satisfies $(1 + \varepsilon)$-perturbation stability, then it has one-versus-all margin $\gamma \geq \frac{\varepsilon^2}{2(\varepsilon+2)}$.*

Consider any cluster $C_i$ with center $c_i$. We must show that any $y \in C_j$ with $j \neq i$ satisfies $d(y, x) > \frac{(\alpha-1)^2}{2(\alpha+1)} \phi_d(C_i)$ for all $x \in C_i$. Let $x' = \arg\max_{x \in C_i} d(x, c_i)$. Clearly, if $d(x', c_i) = 0$ then all points of $C_i$ coincide and $\phi_d(C_i) = 0$. If this is the case, then by the $\alpha$-center proximity, for any $x \in C_i$ and any $y \in C_j$ we have $d(y, x) = d(y, c_i) > \alpha \, d(y, c_j) \geq 0$. Therefore $d(y, x) > a \, \phi_d(C_i)$ for any $a > 0$, which proves our claim.

Suppose instead that $d(x', c_i) > 0$. Choose any $x \in C_i$ and any $y \in C_j$. Now, we have two cases.

**Case 1:** $x = c_i$. In this case we proceed as follows. Bear in mind that $d(x', c_i) = \phi_d(C_i)$, $d(y, c_i) = d(y, x)$, and $d(y, c_j) < \frac{1}{\alpha}d(y, c_i)$.

$$\phi_d(C_i) = d(x', c_i) \tag{25}$$

$$< \frac{1}{\alpha}d(x', c_j) \tag{26}$$

$$\leq \frac{1}{\alpha}\Big(d(x', c_i) + d(c_i, y) + d(y, c_j)\Big) \tag{27}$$

$$< \frac{1}{\alpha}\left(\phi_d(C_i) + d(y, x) + \frac{1}{\alpha}d(y, c_i)\right) \tag{28}$$

$$= \frac{1}{\alpha}\phi_d(C_i) + \frac{1}{\alpha}d(y, x) + \frac{1}{\alpha^2}d(y, x) \tag{29}$$

from which we infer

$$d(y, x) > \frac{\alpha(\alpha - 1)}{\alpha + 1}\phi_d(C_i) > \frac{(\alpha - 1)^2}{2(\alpha + 1)}\phi_d(C_i) \tag{30}$$

**Case 2:** $x \neq c_i$. In this case, $d(x, c_i) > 0$, and we start by deriving:

$$d(y, x) \geq d(x, c_j) - d(y, c_j) \tag{31}$$

$$> d(x, c_j) - \frac{1}{\alpha}d(y, c_i) \tag{32}$$

$$\geq d(x, c_j) - \frac{1}{\alpha}\Big(d(y, x) + d(x, c_i)\Big) \tag{33}$$

$$= -\frac{1}{\alpha}d(y, x) + d(x, c_j) - \frac{1}{\alpha}d(x, c_i) \tag{34}$$

and thus

$$d(y, x)\left(\frac{\alpha + 1}{\alpha}\right) > d(x, c_j) - \frac{1}{\alpha}d(x, c_i) \tag{35}$$

which yields

$$d(y, x) > \frac{\alpha}{\alpha + 1}d(x, c_j) - \frac{1}{\alpha + 1}d(x, c_i) \tag{36}$$

Let $\beta = \frac{d(x', c_i)}{d(x, c_i)}$. Observe that $\phi_d(C_i) \leq 2\beta d(x, c_i)$ and therefore $d(x, c_i) \geq \frac{\phi_d(C_i)}{2\beta}$.

Now we consider two cases. First, suppose that $\beta \leq \frac{\alpha + 1}{\alpha - 1}$. In this case, we apply $d(x, c_j) > \alpha d(x, c_i)$ to (36) to obtain:

$$d(y, x) > \frac{\alpha^2}{\alpha + 1}d(x, c_i) - \frac{1}{\alpha + 1}d(x, c_i) \tag{37}$$

$$= d(x, c_i)(\alpha - 1) \tag{38}$$

$$\geq \frac{\phi_d(C_i)}{2\beta}(\alpha - 1) \tag{39}$$

$$\geq \phi_d(C_i)\frac{(\alpha - 1)^2}{2(\alpha + 1)} \tag{40}$$

Suppose instead that $\beta > \frac{\alpha + 1}{\alpha - 1}$. Since we chose $x' \in C_i$ such that $d(x', c_i) = \beta d(x, c_i)$, we obtain:

$$d(x, c_j) > d(x', c_j) - d(x, x') \tag{41}$$

$$> \alpha d(x', c_i) - \Big(d(x, c_i) + d(x', c_i)\Big) \tag{42}$$

$$= \alpha\beta d(x, c_i) - \Big(d(x, c_i) + \beta d(x, c_i)\Big) \tag{43}$$

$$= d(x, c_i)((\alpha - 1)\beta - 1) \tag{44}$$

Combining this with (36), we obtain:

$$d(y, x) > d(x, c_i) \left( \frac{\alpha}{\alpha + 1}((\alpha - 1)\beta - 1) - \frac{1}{\alpha + 1} \right) \tag{45}$$

$$= d(x, c_i) \left( \frac{\alpha(\alpha - 1)\beta}{\alpha + 1} - 1 \right) \tag{46}$$

$$\geq \frac{\phi_d(C_i)}{2\beta} \left( \frac{\alpha(\alpha - 1)\beta}{\alpha + 1} - 1 \right) \tag{47}$$

$$= \phi_d(C_i) \left( \frac{\alpha(\alpha - 1)}{2(\alpha + 1)} - \frac{1}{2\beta} \right) \tag{48}$$

$$> \phi_d(C_i) \left( \frac{\alpha(\alpha - 1) - (\alpha - 1)}{2(\alpha + 1)} \right) \tag{49}$$

$$= \phi_d(C_i) \frac{(\alpha - 1)^2}{2(\alpha + 1)} \tag{50}$$

Hence, in all cases, we obtain $d(y, x) > \phi_d(C_i) \frac{(\alpha - 1)^2}{2(\alpha + 1)}$. This concludes the proof.

## 5    Proof of Lemma 3

**Lemma 3** (One-versus-all margin implies one-sided-error learnability). *Let $d$ be any pseudometric over $\mathcal{X}$. For any finite $X \subset \mathcal{X}$ and any $\gamma > 0$, define the effective concept class over $X$:*

$$H = \{C \subseteq X \: : \: d(X \setminus C, C) > \gamma \, \phi_d(C)\} \tag{2}$$

*Then $H = I(H)$, and $\mathrm{vc\text{-}dim}(H, X) \leq M^*(\gamma, d)$ where $M^*(\gamma, d) = \max(2, M(\gamma, d))$. Therefore, $H$ can be learned with one-sided error $\varepsilon$ and confidence $\delta$ with $\mathcal{O}\big(\varepsilon^{-2}(M^*(\gamma, d) \log 1/\varepsilon + 1/\delta)\big)$ examples by choosing the smallest consistent hypothesis in $H$.*

For the first claim, we start by showing that $H = I(H)$. Let $C_1, C_2 \in H$. We show that $C := C_1 \cap C_2 \in H$, too. Consider any $y \in X \setminus C$, and without loss of generality assume that $y \notin C_1$. Since $C \subseteq C_1$, we have $\phi_d(C_1) \geq \phi_d(C)$ and $d(y, C) \geq d(y, C_1)$. Moreover, $d(y, C_1) > \gamma \, \phi_d(C_1)$ since $C_1 \in H$. Therefore:

$$d(y, C) \geq d(y, C_1) > \gamma \, \phi_d(C_1) \geq \gamma \, \phi_d(C) \tag{51}$$

proving that $d(y, C) > \gamma \, \phi_d(C)$. Since this holds for all $y \in X \setminus C$, we have $C \in H$ as well. Therefore, $H = I(H)$.

Since $H = I(H)$, then $\mathrm{vc\text{-}dim}(H, X) = \mathrm{vc\text{-}dim}(I(H), X)$. Now, we use the following results of Kivinen [1995]:

**Definition 8.** *[Kivinen [1995], Definition 5.11] Let $\mathcal{X}$ be any domain and $\mathcal{H} \subseteq 2^{\mathcal{X}}$ be a concept class. We say that $\mathcal{H}$ slices $X \subset \mathcal{X}$ if, for each $x \in X$, there is $h \in \mathcal{H}$ such that $X \cap h = X \setminus \{x\}$. The slicing dimension of $(\mathcal{H}, \mathcal{X})$, denoted by $\mathrm{sl}(\mathcal{H}, \mathcal{X})$, is the maximum size of a set sliced by $\mathcal{H}$. If $\mathcal{H}$ slices arbitrarily large sets, then we let $\mathrm{sl}(\mathcal{H}, \mathcal{X}) = \infty$.*

**Lemma 11** (Kivinen [1995], Lemma 5.19). *If $\mathrm{sl}(\mathcal{H}, \mathcal{X}) < \infty$, then $\mathrm{vc\text{-}dim}(I(\mathcal{H}), \mathcal{X}) \leq \mathrm{sl}(\mathcal{H}, \mathcal{X})$.*

We will now show that $\mathrm{sl}(H, X) \leq M^*(\gamma, d)$, where $M^*(\gamma, d)$ is finite by assumption. By Lemma 11, this will imply $\mathrm{vc\text{-}dim}(H, X) \leq \mathrm{sl}(H, X)$.

Let $S \subseteq X$ be any set sliced by $H$. We show that $|S| \leq M^*(\gamma, d)$. The case $|S| \leq 2$ is trivial since $M^*(\gamma, d) \geq 2$. Suppose then that $|S| \geq 3$. Choose $a, b \in S$ such that $d(a, b) = \phi_d(S)$. Since $S$ is sliced by $H$, for any $x \in S$ we have $S \setminus x = S \cap C$ for some $C \in H$. Since $S \setminus x \subseteq C$, we have $d(S \setminus x, x) \geq d(C, x)$ and $\phi_d(C) \geq \phi_d(S \setminus x)$. Moreover, $d(C, x) > \gamma \, \phi_d(C)$, since $x \in X \setminus C$ and $C \in H$. Therefore:

$$d(S \setminus x, x) \geq d(C, x) \tag{52}$$

$$> \gamma \, \phi_d(C) \tag{53}$$

$$\geq \gamma \, \phi_d(S \setminus x) \tag{54}$$

Hence, $d(S \setminus x, x) > \gamma \phi_d(S \setminus x)$ for all $x \in S$.

Now, suppose first that $\gamma \geq 1$. Since $|S| \geq 3$, any $x \in S \setminus \{a, b\}$ yields the absurd:

$$\phi_d(S) \geq d(S \setminus x, x) \tag{55}$$
$$> \phi_d(S \setminus x) \qquad \text{since } \gamma \geq 1 \tag{56}$$
$$= d(a, b) \qquad \text{since } a, b \in S \setminus x \tag{57}$$
$$= \phi_d(S) \qquad \text{by the choice of } a, b \tag{58}$$

Hence we must have $|S| \leq 2$, which implies $|S| \leq M^*(\gamma, d)$. Suppose instead that $\gamma < 1$. Then, for any two distinct points $x, y \in S$, we have $d(x, y) > \gamma \phi_d(S)$. This is trivially true if $x = a$ and $y = b$; otherwise, assuming $x \notin \{a, b\}$, it follows by the fact that $d(x, y) \geq d(S \setminus x, x) > \gamma \phi_d(S \setminus x) = \gamma \phi_d(S)$, as seen above. Moreover, $S$ is contained in the closed ball $B(x, \phi_d(S))$ for any $x \in S$. Therefore, $\mathcal{M}(B(x, r), \gamma r, d) \geq |S|$ for $r = \phi_d(S)$ and some $x \in \mathcal{X}$. By definition of $M(\gamma, d)$ this implies that $|S| \leq M(\gamma, d)$, and $M(\gamma, d) \leq M^*(\gamma, d)$. This concludes the proof.

# 6 Proof of Theorem 3

**Theorem 3.** *Let $(X, O)$ be any instance whose latent clustering $\mathcal{C}$ has one-versus-all margin $\gamma > 0$ with respect to $d_1, \ldots, d_k$. Then $\mathrm{MREC}(X, O, \gamma, d_1, \ldots, d_k)$ outputs $\mathcal{C}$ while making, with high probability, at most $\mathcal{O}(M^*(\gamma) k \log k \log n)$ label queries to $O$, where $M^*(\gamma) = \max(2, M(\gamma))$. Moreover, for any algorithm $\mathcal{A}$ and for any $\gamma > 0$, there are instances with one-versus-all margin $\gamma$ on which $\mathcal{A}$ makes $\Omega(M(2\gamma))$ label queries in expectation.*

For the lower bounds, let $\gamma' = 2\gamma$. By definition of $M(\gamma')$, there exists a set $X$ of $M(\gamma')$ points that, according to some $d \in \{d_1, \ldots, d_k\}$, lies within a ball of radius $r > 0$, and thus has diameter at most $2r$, and such that $d(x, y) > \gamma' r = 2\gamma r$ for all distinct $x, y \in X$. Now choose $x \in X$ uniformly at random, and define $\mathcal{C} = (x, X \setminus x)$. The argument above shows that $d(x, X \setminus x) > \gamma \phi_d(X)$, which implies that $\mathcal{C}$ has one-versus-all margin $\gamma$. Clearly, in expectation over the distribution of $\mathcal{C}$, any exact cluster recovery algorithm must make $\Omega(|X|) = \Omega(M(2\gamma))$ queries. By Yao's principle for Monte Carlo algorithms, then, any such algorithm makes $\Omega(M(2\gamma))$ queries on some instance.

Let us turn to the upper bounds. The pseudocode of $\mathrm{MREC}$ is given below. As in the proof sketch, for each $i \in [k]$ the class $H_i$ is defined as:

$$H_i = \{C \subseteq X \ : \ d_i(X \setminus C, C) > \gamma \phi_{d_i}(C)\} \tag{59}$$

By Lemma 3 and by definition of learning with one-sided error (Definition 1), we have $\widehat{C}_i \subseteq C_i$ and therefore $\mathrm{MREC}$ never misclassifies any point, and moreover:

$$\mathbb{P}\left(\left|C_i \setminus \widehat{C}_i\right| \leq \varepsilon |X|\right) \geq 1 - \delta \tag{60}$$

In our case, that is, with $\varepsilon = 1/2k$ and $\delta = 1/2$, and since $|C_i \setminus \widehat{C}_i| = |C_i| - |\widehat{C}_i|$, this yields:

$$\mathbb{P}\left(\left|\widehat{C}_i\right| \geq |C_i| - \frac{|X|}{2k}\right) \geq \frac{1}{2} \tag{61}$$

which, since $\left|\widehat{C}_i\right| \geq 0$, implies:

$$\mathbb{E}\left|\widehat{C}_i\right| \geq \frac{1}{2} \cdot \left(|C_i| - \frac{|X|}{2k}\right) = \frac{|C_i|}{2} - \frac{|X|}{4k} \tag{62}$$

By summing over all $i \in [k]$, at each round $\mathrm{MREC}$ correctly labels, and removes from $X$, an expected number of points equal to:

$$\mathbb{E}\left|\widehat{C}_1 \cup \ldots \cup \widehat{C}_k\right| = \sum_{i=1}^{k} \mathbb{E}\left|\widehat{C}_i\right| \geq \sum_{i=1}^{k}\left(\frac{|C_i|}{2} - \frac{|X|}{4k}\right) = \frac{|X|}{2} - \frac{|X|}{4} = \frac{|X|}{4} \tag{63}$$

Thus, at each round $\mathrm{MREC}$ gets rid of an expected fraction $1/4$ of all points still in $X$. By a standard probabilistic argument this implies that, with high probability, all points are correctly labeled within $\mathcal{O}(\log n)$ rounds, see Lemma 3 of [Bressan et al., 2020]. Therefore, the total number of queries is bounded by $\mathcal{O}(M^*(\gamma) k \log k)$ with high probability. This concludes the proof.

**Algorithm 3** MREC$(X, O, \gamma, d_1, \ldots, d_k)$

---

**if** $X = \emptyset$ **then return**
for each $i \in [k]$ let $H_i$ as in Equation 59
draw a sample $S$ of $|S| = \Theta\big(M^*(\gamma) k \ln k\big)$ points u.a.r. from $X$
use $O$ to learn the labels of $S$
**for** $i \in [k]$ **do**
    let $S_i$ be the subset of $S$ having label $i$
    let $\widehat{C}_i$ be the smallest set in $H_i$ consistent with $S_i$
    give label $i$ to every $x \in \widehat{C}_i$
let $X' = X \setminus \cup_{i \in [k]} \widehat{C}_i$
MREC$(X', O, \gamma, d_1, \ldots, d_k)$

---

# 7 Proof of Theorem 4

As said in the sketch, the proof follows the same ideas of the proof of Theorem 3.

For the lower bounds, suppose that $\mathrm{cosl}(\mathcal{H}) < \infty$, and let $X$ be a set cosliced by $\mathcal{H}$ with $|X| = \mathrm{cosl}(\mathcal{H})$. We draw a random uniform element $x \in X$, and we consider the clustering $\mathcal{C} = (x, X \setminus x)$. By definition of sliced set, $\mathcal{C}$ is realised by $\mathcal{H}$, and therefore it satisfies the assumptions. Now the same arguments of the lower bounds of Theorem 3 imply that any algorithm needs $\Omega(|X|) = \Omega(\mathrm{cosl}(\mathcal{H}))$ queries on some instance to return $\mathcal{C}$. Clearly, if $\mathrm{cosl}(\mathcal{H}) = \infty$ this means that we can take $|X| = n$ arbitrarily large, whence the second lower bound.

For the upper bounds, let $P_k(X)$ be the set of all $k$-clusterings of $X$ that are realised by $\mathcal{H}$. Then, for each $i \in [k]$ we define:

$$H_i = \{C' \,:\, C' = C_i' \wedge (C_1', \ldots, C_k') \in P_k(X)\} \tag{64}$$

As in MREC, we learn each class $H_i$ with one-sided error by choosing the smallest hypothesis in $I(H_i)$ that is consistent with $S_i$, for a labeled sample $S$ of size $\Theta(\text{vc-dim}(I(H_i), X) \, k \ln k)$. As shown in the proof of Lemma 3, a result of Kivinen [1995] implies that if $\mathrm{sl}(H_i, X) < \infty$ then vc-$\dim(I(H_i), X) \leq \mathrm{sl}(H_i, X)$. Therefore, to prove the theorem we only need to show that $\mathrm{sl}(H_i, X) \leq \mathrm{cosl}(\mathcal{H})$. To this end, suppose that $U = \{x_1, \ldots, x_\ell\} \subseteq X$ is sliced by $H_i$. By construction of $H_i$, this means that there are $\ell$ clusterings $\mathcal{C}_1, \ldots, \mathcal{C}_\ell$, each one realised by $\mathcal{H}$, such that $\mathcal{C}_i = (\{x_i\}, U \setminus \{x_i\})$ for all $i \in [k]$. This implies that $U$ is cosliced by $\mathcal{H}$. Hence, $|U| \leq \mathrm{cosl}(\mathcal{H})$ and so $\mathrm{sl}(H_i, X) \leq \mathrm{cosl}(\mathcal{H})$, as claimed. The rest of the proof is similar to the proof of Theorem 3, and shows that at each round we recover an expected constant fraction of all points, and that therefore all points will be recovered with high probability after $\mathcal{O}(\log n)$ rounds. This shows that we can recover $\mathcal{C}$ by making with high probability at most $\mathcal{O}(\mathrm{cosl}(\mathcal{H}) k \log k \log n)$ queries.

# 8 Proof of Theorem 5

**Theorem 5.** *Let $\mathcal{H}$ be a concept class in $\mathbb{R}^m$ that is non-fractal and closed under affine transformations. There is an algorithm that, given any instance whose latent clustering $\mathcal{C}$ has one-versus-all margin $\gamma$ and is realized by $\mathcal{H}$, returns $\mathcal{C}$ while making $\mathcal{O}(Mk \log k \log n)$ label queries with high probability, where $M = \max\big(2, (1 + 4/\gamma)^m\big)$. Moreover, for any algorithm $\mathcal{A}$, there exist arbitrarily large $n$-point instances, whose latent clustering $\mathcal{C}$ has arbitrarily small one-versus-all margin and is realized by $\mathcal{H}$, where $\mathcal{A}$ makes $\Omega(n)$ label queries in expectation to recover $\mathcal{C}$.*

The upper bound follows from Theorem 3, by using the standard fact that in the Euclidean metric the unit ball has covering number $\mathcal{N}(B(x, 1), \varepsilon) \leq (1 + 2/\varepsilon)^m$ for all $\varepsilon > 0$. As it is well-known, we have $\mathcal{M}(B(x, 1), \varepsilon) \leq \mathcal{N}(B(x, 1), \varepsilon/2)$, which for $\varepsilon = \gamma$ yields $M(\gamma) \leq (1 + 4/\gamma)^m$.

We now prove the lower bound. Having instances with "arbitrarily small one-versus-all margin" means that $\gamma = 0$, see Definition 3. Take any $h \in \mathcal{H}$ such that both $h$ and its complement $\bar{h}$ contain a ball of positive radius. Note that this implies that, for any $\rho > 0$, there exists a closed ball $B$ with

radius $r > 0$ such that:

$$B \subseteq h \tag{65}$$

$$\exists x \in \overline{h} \ : \ d(B, x) \leq \rho \tag{66}$$

Let $c$ be the center of $B$. Consider a sphere $S$ of radius $r'$ that contains $x$ and whose center $c'$ lies on the affine subspace $x + \alpha(c - x)$. Let $\eta = \sup_{y \in S \setminus B} d(x, y)$ and let $X$ be an $\eta$-packing of $S$. Note that, since $\gamma = 0$, we can choose $\rho$ and $r'$ arbitrarily small, and in particular we can make the ratio $\frac{\eta}{r'}$ arbitrarily small. This implies that we can make $X$ arbitrarily large, see Figure 4.

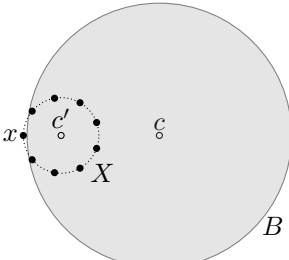

Figure 4: An $\eta$-packing $X$ such that $X \cap h = X \setminus \{x\}$ and $X \cap \overline{h} = \{x\}$. The ball $B$ is by construction entirely in $h$, whereas $x$ is by construction in $\overline{h}$.

Now, consider $x' \in X \setminus \{x\}$. As by construction $d(x', x) > \eta$, and as $r' < r$, we must have $x' \in B$. Therefore, $x' \in h$. Hence the concept $h_x = h$ is such that $X \cap h_x = X \setminus \{x\}$. Now, for any $x' \in X \setminus \{x\}$, there is a rotation $R$ with fixed point $c'$ and such that $R(x') = x$. Hence, $h_{x'} = R^{-1} h_x$ is such that $X \cap h_{x'} = X \setminus \{x'\}$. Since $R^{-1}$ is an affine transformation, $h_{x'} \in \mathcal{H}$ as well. Hence, for every $x \in X$ there exists some concept $h_x \in \mathcal{H}$ such that $X \cap h_x = X \setminus \{x\}$.

Now consider the complement $\overline{h}$ of $h$. Note that $\overline{h} \in \text{co}(\mathcal{H})$. The first part of the argument above can be applied to $\overline{h}$ as well, showing that for every $x \in X$ there exists $\overline{h_x} \in \text{co}(\mathcal{H})$ such that $X \cap \overline{h_x} = X \setminus \{x\}$. Now consider the complement $h_x$ of $\overline{h_x}$. Clearly $h_x \in \mathcal{H}$, and moreover, $X \cap h_x = \{x\}$.

Hence, for any $x \in X$ there are two concepts $h_x^-, h_x^+ \in \mathcal{H}$ such that $X \cap h_x^- = X \setminus \{x\}$ and $X \cap h_x^+ = \{x\}$. Hence, every 2-clustering $\mathcal{C}$ of $X$ in the form $C_1 = \{x\}, C_2 = X \setminus \{x\}$ is realized by $\mathcal{H}$. Note that this holds with $X$ fixed; we just need to transform the concetps appropriately. It is they immediate to see that any algorithm must perform $\Omega(|X|)$ queries on some instance $(X, O)$. As we can make $X$ arbitrarily large, this completes the proof.