# OpenReview forum: "On Margin-Based Cluster Recovery with Oracle Queries"
_NeurIPS.cc/2021/Conference — NeurIPS 2021 Poster_

### Official Review · Reviewer_jtkL · 2021-07-07

**Rating:** 7
**Confidence:** 3

**Summary:**

This work studies query efficient cluster recovery with label queries. In particular, given a k-clustering $(C_1,\ldots,C_k)$ of an $n$ point set $X$, the learner may ask, adaptively, for any $x \in X$, “which cluster contains x?” The goal is to recover the clustering on all n points using only $O_k(\log(n))$ queries. This is a very natural problem, variants of which have seen significant study both within the clustering (e.g. Ashtiani et al. 2016) and active learning literature (e.g. Kane et al. 2017).

The authors prove a number of interesting results to this effect. In the Euclidean setting, the authors introduce a novel complexity measure called the “Convex Hull Margin” that non-trivially generalizes previous sufficient conditions for query efficient cluster recovery such ball and ellipsoid-based margins and the SVM margin. They show that small convex hull margin implies computational and query efficient cluster recovery (with near optimal query complexity), even improving previous special cases such as the ellipsoid margin. The authors also study an extension to non-Euclidean spaces which are only assumed to have some fixed set of pseudometrics. In this case, they introduce a novel measure called “one-versus-all” margin. They prove this measure generalizes a number of known cluster recovery results such as center proximity and perturbation stability, and also gives a sufficient condition for query (but not computationally) efficient cluster recovery.

Finally, the authors also address an interesting variant of cluster recovery where the clusters must be realized by some fixed concept class H. In this scenario, the authors introduce coslicing dimension which captures a well-known lower bound of Dasgupta showing that efficient learning is impossible when every point can be cutoff from the rest. They then proceed to show that this is in fact the only barrier to efficient cluster recovery—proving that finite coslicing dimension is necessary and sufficient.

**Main Review:**

This work makes significant progress on a natural, important problem in both clustering and active learning. The results are very clean, and likely of broad interest to the learning theory community. The techniques, while relatively simple, are an interesting mix of combinatorial, probabilistic, and inference based methods that could easily be of use in future work. On top of this the paper is well-written, and has careful analysis of their relation to margin parameters in prior work. The proofs I checked seemed sound, and no outlandish claims are made otherwise.

My only real complaints are due to a few missing references.

First, the “Convex Hull Expansion Trick” is not novel. The same trick was introduced by Hopkins, Kane, Lovett, and Mahajan (COLT 2020) to bound the inference dimension of clusters (when considering halfspaces under TNC noise). However their analysis recovers a slightly worse query complexity bound, and may be of narrower scope than the general pseudometric variant considered in this work. The analysis is also quite different, as HKLM rely on LP duality.

Second, the coslicing dimension lower bound generalizes a result due to Dasgupta (in “Analysis of a greedy active learning strategy”). This should be mentioned. In fact I would say it is a strength of the result, the authors show that Dasgupta’s classic bound is essentially the only barrier to cluster recovery in this setting!

Finally a few minor notes:

Usually query complexity is included in computational complexity, so it doesn’t make much sense to say your algorithm uses exp(m) queries but runs in poly(m) time. Plus, the regime you consider is only really interesting for n >> 2^m (otherwise you can’t do inference), so poly(m) vs exp(m) runtime doesn’t really make a difference anyway.

I think the terminology “outputs C deterministically" is a bit non-standard (though not technically wrong) given that the algorithm is randomized. Usually this would be referred to as a “randomized zero-error” algorithm.

**Time Spent Reviewing:**

8

---

> ### Author Response · Authors · 2021-08-09
> **Author response**
>
> _First, the “Convex Hull Expansion Trick” is not novel. The same trick was introduced by Hopkins, Kane, Lovett, and Mahajan (COLT 2020) to bound the inference dimension of clusters (when considering halfspaces under TNC noise). Their analysis recovers a slightly better query complexity bound, but may be of narrower scope than the general pseudometric variant considered in this work. The analysis is also quite different, as HKLM rely on LP duality._
>
> We respectfully disagree with the reviewer’s claim. Hopkins et al. show that the volume of the convex hull of the learned points increases by a constant factor every time we include a new point whose label could not be inferred from the previous points. Here we show that, by actively rescaling the convex hull of the sample about its center of mass, we recover a constant fraction of the cluster without any mistake. As far as we can see, the two results are different and one does not imply the other, as confirmed by the fact that the two proofs are significantly different. We agree however that both results are based on the margin-based analysis of volumes of convex bodies, and therefore may appear similar at first sight.
>
>
> _Second, the coslicing dimension lower bound generalizes a result due to Dasgupta (in “Analysis of a greedy active learning strategy”). This should be mentioned. In fact I would say it is a strength of the result, the authors show that Dasgupta’s classic bound is essentially the only barrier to cluster recovery in this setting._
>
> Thanks for your comment. Indeed Kivinen’s slicing dimension and Hanneke’s star number, which we mention, build on Dasgupta’s example. We will make this connection more explicit in the revision.
>
> _Usually query complexity is included in computational complexity, so it doesn’t make much sense to say your algorithm uses $\exp(m)$ queries but runs in $poly(m)$ time. Plus, the regime you consider is only really interesting for $n \gg 2^m$ (otherwise you can’t do inference), so $poly(m)$ vs $\exp(m)$ runtime doesn’t really make a difference anyway._
>
> Our query complexity is, to be precise, bounded by the minimum between $n=|X|$ and the $\exp(m) \log(n)$ term. However, in our bounds we showed just the $\exp(m) \log(n)$ term in order to highlight the dependence on $\log(n)$, which is the interesting part. We agree however that this can be misleading, and we will clarify this point.
>
> _I think the terminology “outputs C deterministically" is a bit non-standard (though not technically wrong) given that the algorithm is randomized. Usually this would be referred to as a “randomized zero-error” algorithm._
>
> Thank you, we will fix this.

---

> > ### Comment · Reviewer_jtkL · 2021-08-10
> > **Reviewer Response**
> >
> > Author Comment: "We respectfully disagree with the reviewer’s claim. Hopkins et al. show that the volume of the convex hull of the learned points increases by a constant factor every time we include a new point whose label could not be inferred from the previous points. Here we show that, by actively rescaling the convex hull of the sample about its center of mass, we recover a constant fraction of the cluster without any mistake. As far as we can see, the two results are different and one does not imply the other, as confirmed by the fact that the two proofs are significantly different. We agree however that both results are based on the margin-based analysis of volumes of convex bodies, and therefore may appear similar at first sight."
> >
> > Reviewer Response: This is not quite the full picture. In particular, the entire point of HKLM's lemma is exactly that if a random sample of $\tilde{O}((1+1/\gamma)^m)$ points are drawn from a cluster $C$, then in expectation the $(1+\gamma)$-scaling of the convex hull of the sample contains a constant fraction of $C$ (and thus can learn a constant fraction of $C$ with no mistakes). This is an immediate corollary of HKLM's lemma and standard inference dimension arguments [KLMZ17], and is how HKLM use the result in the rest of their work.
> >
> > On the other hand, it is true that the exact method of scaling in this work differs from HKLM, and that HKLM is not given for general pseudometrics. Regardless, since HKLM do prove that expanding the convex hull by $(1+\gamma)$ infers a constant fraction of $C$, I still believe that calling the convex hull expansion trick new is misleading. If the authors wish to claim this, they need to clarify that the novel component is the new type of scaling and generality for pseudometrics. As it is currently introduced in the paper (e.g. line 55-57), the technique seems basically identical to HKLM.

---

> > > ### Author Response · Authors · 2021-08-12
> > > **Author Response**
> > >
> > > Thanks for your response. We would like to summarize our view as follows.
> > >
> > > First, we agree with you that the general idea of rescaling the convex hull of the sample by some multiplicative factor is not novel. It was not our intention to claim novelty for that, and we will make sure to avoid any ambiguities in the revision.
> > >
> > > Second, we doublechecked HKLM, in particular Lemma 34 and its proof, and we cannot see how our Lemma 1 can be obtained as a simple corollary of that Lemma. We are aware of inference dimension techniques, and we agree that, as far as the query complexity part of our expansion trick is concerned, it would suffice to prove that the inference dimension of clusters with a convex hull margin of $\gamma$ is roughly $O(1+1/\gamma)^m$. We would not be surprised if there were a short alternative proof for that, but we fail to see how such a proof can be obtained from Lemma 34 of HKLM, which (translated in our terms) assumes $\gamma=m$ and, for this case, proves an inference dimension bound of $O(m \log m)$. (Incidentally, and with the only purpose of saying that this case is not new to us, when $\gamma=m$ one can obtain a query complexity bound of $m^2$ by taking the MVE of $S$, a.k.a. the John’s ellipsoid, which is worse than HKLM’s $O(m \log m)$ but still polynomial). We also note that, when $\gamma=\Theta(m)$, our result gives a query complexity bound of $poly(m)$, thus agreeing with Lemma 34 (although with significantly higher bounds).
> > >
> > > In any case, we are happy to add a discussion of the connections with HKLM, clarifying our elements of novelty. We agree that there are conceptual overlaps between the two works and we thank you for bringing this to our attention.

---

> > > > ### Comment · Reviewer_jtkL · 2021-08-13
> > > > **Reviewer Response**
> > > >
> > > > Thank you for the informative response. I trust the authors to decide how HKLM relates to their work, and am happy to recommend acceptance of their paper regardless.
> > > >
> > > > I will add one clarification, which is that HKLM's proof does not really rely on the assumption $\gamma = m$. One can replace appropriate occurrences of $m$ in their proof with $\gamma$, and the same arguments go through. This results in the following statement:
> > > >
> > > >
> > > > Let $(C_1,\ldots, C_k)$ be a family of clusters such that for all $C_i=${$x_1,\ldots,x_r$} and $y \in C_j$ (for $i \neq j$), $y$ $\textbf{cannot}$ be expressed as an affine combination $y=\sum\limits_{\ell=1}^r \alpha_\ell x_\ell$ satisfying
> > > > $\sum\limits_{\ell=1}^r |\alpha_\ell| \leq 1+\gamma$. Then the inference dimension of this family is at most $\text{poly}(m) \cdot (1+\frac{2}{\gamma})^m$ (notably worse by a factor of $2^m$ over the bound in this work--I will correct this in the main review).

---

### Official Review · Reviewer_fefH · 2021-07-17

**Rating:** 6
**Confidence:** 3

**Summary:**

This work examines a cluster recovery problem where the algorithm has access to an oracle that can return whether two points lie in the same cluster. It builds on a line of work [Ashtiani et al. ‘16] [Bressan et al. ‘20] that does so with only O(log n) queries but under a margin property (the dependence on the margin is hidden in the number of queries). [Ashtiani et al. ‘16] did so with a spherical margin property: there is a sphere around each cluster separating it from all other points, and this had to continue to hold as the sphere was expanded by a (1 + gamma) factor. [Bressan et al. ‘20] generalized this assumption to ellipsoidal margins, and this work further generalizes that to “convex hull” margins and even further to “one-versus-all” margins.

**Limitations And Societal Impact:**

An optimistic view of where this line of research leads might be that future work improves the other factors in the query complexity / running time and it winds up being more applicable. I can see this work fitting into this hypothetical overall progression nicely due to the new margins and techniques it introduces.

**Main Review:**

Although all three of [Ashtiani et al. ‘16], [Bressan et al. ‘20], and this work manage to only use O(log n) queries, this way of writing the bound tunnel visions on the dependence on the number of points n and ignores the number of clusters k and the margin gamma. I went through the other papers, and the queries seem to have the following dependence on these other two parameters (omitting the dependence on the algorithm failure probability delta) (translating to the variable names used by this paper):
- [Ashtiani et al. ‘16]: O(k log n + k^2 / gamma^4 log k)
- [Bressan et al. ‘20]: O(k log n (k^2 m^2 log k + 2^m (m / gamma log (m / gamma))^m)
- This work: O(k^2 m^5 (1 + 1/gamma)^m log (1 + 1/gamma) log n)

Notably, with respect to the dimension m, this work has better dependence compared to [Bressan et al.’ 20] (which the authors point out in the paper) but worse dependence compared to [Ashtiani et al. ‘16]. In some sense then, this work is an improvement to the ellipsoid margin case, which seems much simpler to justify the impact of (compared to trying to come up with cases that fall under convex hull margins but not ellipsoid margins); [Bressan et al. ‘20] point out ellipsoids make sense over spheres when one expects the features to be on different scales.

For both of these last two results, I am concerned that the appearance of m in the exponent makes them not that useful. In the experimental section of [Bressan et al. ‘20], the problem dimension is chosen to be quite small: 2, 4, 6, or 8. This paper is missing an experimental section, and so the question of how the proposed algorithm practically compares to its two predecessors is left open.

It’s also worth considering exactly how much the generalized margin properties allow one to capture additional interesting cases. The paper proves that several other properties imply the one-versus-all margin, including the SVM margin, alpha-center proximity, and (1 + eps)-perturbation resilience.

Overall, I think this result can be viewed as a nice theoretical contribution on past work. I wish the practical consequences were given a closer look, but I’d still lean towards acceptance to NeurIPS.

Misc.:
- Line 243: Equation (2) does not appear to be centered properly.

-----
I've read the author's response, thank you!


**Time Spent Reviewing:**

3 hours

---

> ### Author Response · Authors · 2021-08-09
> **Author response**
>
> About the dependence on other parameters: Thanks for your comment. Although the dependence on $\gamma$ and $m$ for all three algorithms was already mentioned in the introduction, we will follow your suggestion and highlight it even better.
>
> About the experiments: We agree that an experimental comparison would be interesting. However, we decided to keep this work purely theoretical, and leave experiments for future research.

---

### Official Review · Reviewer_BX75 · 2021-07-19

**Rating:** 7
**Confidence:** 4

**Summary:**

This paper considers the problem of finding convex clusters in space using a small number of cluster queries. It gives an algorithm for this problem, and also extends the algorithm to the one-vs-all margin. It further shows a connection to exact active cluster recoverability.

**Limitations And Societal Impact:**

Yes.

**Main Review:**

The first problem considered is where a label is defined by a convex body with a margin, such that if the convex body is expanded outwards by a small factor times its diameter, it does not intersect any new points. The algorithm to find a cluster samples points, draws a convex body around those of the same label, and then grows the body by a constant factor, about the center of mass of the points. The authors show with high probability the new body either contains a significant number of new points of the same label (or contains all points of the label). This is done by combining two separate combinatorical results, the first by Naszodi which bounds the number of vertices of the body, and the second a recent breakthrough of Kupavskii which gives the VC-dimension of this object. While the overview is quite straightforward, the details are not.

The authors then consider more general spaces, and define a one-vs-all margin similar to above, but without the requirements of Euclidean space, or even convexity. They show that it generalizes the clustering notions of center proximity and perturbation stability. They then demonstrate that such a margin implies learnability, and show how to plug it in to an analogue of the previous algorithm, to again learn clusters using a small number of label queries. This demonstrates that their framework can be applied to a range of setting.

I found the results in this paper to be interesting and also deep while still being accessible. I would anticipate them being applied to other learning problems in related fields as well.

Minor comments:
I think the authors should expand on the comment in line 245 on fixing the pseudometrics in advance, and how this differs from what is done in line 141. I found this confusing.

The abstract and introduction inexplicably seem to think that R^m and Euclidean space are the same thing.

The opening assertion that queries identifying a point's cluster and queries determining whether a pair are in the same cluster are the equivalent, but this is only true if k is small and one a representative of each cluster at hand.

Typos: k-uple should be k-tuple.

**Time Spent Reviewing:**

4 hours

---

> ### Author Response · Authors · 2021-08-09
> **Author response**
>
> _I think the authors should expand on the comment in line 245 on fixing the pseudometrics in advance, and how this differs from what is done in line 141. I found this confusing._
>
> We agree. We will expand on the comment if space allows it.
>
> _The abstract and introduction inexplicably seem to think that $R^m$ and Euclidean space are the same thing._
>
> Thank you for pointing this out. We will use a more precise terminology.
>
> _The opening assertion that queries identifying a point's cluster and queries determining whether a pair are in the same cluster are the equivalent, but this is only true if $k$ is small and one a representative of each cluster at hand._
>
>  We agree, and indeed we explain this in Section 2. In the introduction we opted for a simplified sentence that captures the essence.
>
> _Typos: k-uple should be k-tuple._
>
> Thanks!

---

> > ### Comment · Reviewer_BX75 · 2021-08-25
> > **Rebuttal**
> >
> > I am satisfied with the authors' rebuttal.

---

### Decision · Program_Chairs · 2021-09-27

**Decision:**

Accept (Poster)

**Comment:**

The paper discusses a clustering setting in which clusters are recovered using queries of the form “are two points x, x’ in the same cluster”, in a general notion of margin which generalizes previously studied cases.  The paper is theoretical, with no experimental part.  Reviews are high quality, with consistent pro-accept bottom line.  Reviewers raised some concerns related to novelty of some notions defined in the paper, and also related to lack of experiments, but these do not seem to considerably impact the final opinion.